# Mixture of Experts Made Personalized: Federated Prompt Learning for Vision-Language Models

**Jun Luo**
Intelligent Systems Program
University of Pittsburgh
Pittsburgh, PA 15213, USA
jul117@pitt.edu

**Chen Chen**
Center for Research in Computer Vision
University of Central Florida
Orlando, FL 32816, USA
chen.chen@crcv.ucf.edu

**Shandong Wu**
Intelligent Systems Program
Department of Radiology
Department of Biomedical Informatics
Department of Bioengineering
University of Pittsburgh
Pittsburgh, PA 15213, USA
wus3@upmc.edu

## Abstract

Federated prompt learning benefits federated learning with CLIP-like Vision-Language Model's (VLM's) robust representation learning ability through prompt learning. However, current federated prompt learning methods are habitually restricted to the traditional FL paradigm, where the participating clients are generally only allowed to download a single globally aggregated model from the server. While justifiable for training full-sized models under federated settings, in this work, we argue that this paradigm is ill-suited for lightweight prompts. By facilitating the clients to download multiple pre-aggregated prompts as fixed non-local experts, we propose **P**ersonalized **Fed**erated **M**ixture of **A**daptive **P**rompts (**pFedMoAP**), a novel FL framework that personalizes the prompt learning process through the lens of Mixture of Experts (MoE). pFedMoAP implements a local attention-based gating network that learns to generate enhanced text features for better alignment with local image data, benefiting from both local and downloaded non-local adaptive prompt experts. Extensive experiments on 9 datasets under various federated settings demonstrate the efficacy of the proposed pFedMoAP algorithm. The code is available at https://github.com/ljaiverson/pFedMoAP.

## 1 Introduction

Recent years have witnessed the prosperity of federated learning (FL) (McMahan et al., 2017; Kairouz et al., 2019; Li et al., 2020a) as a potent paradigm for training machine learning models across decentralized data sources. While offering a privacy-preserving solution in collaborative training settings, this approach faces a critical challenge in the form of heterogeneous data distribution across the participating clients (Li et al., 2020a). And due to the data heterogeneity, the generalization capabilities of the trained models may be compromised (Li et al., 2020b; Sahu et al., 2018). In this context, the pre-trained Vision-Language Models (VLMs) such as CLIP (Radford et al., 2021) and ALIGN (Jia et al., 2021) can provide a significant impact with their remarkable applications in learning transferable representations across a wide range of tasks and disparate data distributions encountered in federated settings.

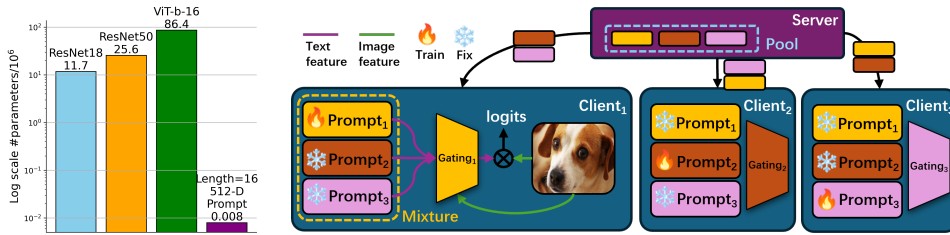

(a) Model parameter counts      (b) Overall pipeline of `pFedMoAP` over three clients

Figure 1: Schematic diagram of employing Mixture of Experts into federated learning. We facilitate the sharing of pre-aggregated prompts thanks to their lightweight nature. Each client downloads the pre-aggregated prompts trained on the remaining two clients through the server, keeping them fixed locally as non-local experts.

However, applying VLMs in FL contexts is not without challenges. Due to their millions of parameters, fine-tuning these large-scale models in federated settings incurs prohibitively high communication overhead, making it impractical for many real-world applications. This limitation has led researchers to explore more efficient adaptation techniques for VLMs, with prompt learning emerging as a promising approach to overcome the communication bottleneck.

Prompt learning for VLMs (Zhou et al., 2022b), originally proposed to eliminate the need for hand-crafted prompts, replaces the context words with small-scale learnable vectors while keeping the pre-trained model fixed. Such a lightweight approach particularly benefits FL by tremendously reducing the communication overhead associated with transmitting the entire fine-tuned model. For instance, `PromptFL` (Guo et al., 2023b) leverages a `FedAvg` (McMahan et al., 2017) style aggregation of the locally trained prompts. `FedPR` (Feng et al., 2023) learns the visual prompt within the null space of the global prompt. In addition, recent criticism on prompt-based VLMs to unseen data distributions (Zhou et al., 2022b; Khattak et al., 2023a;b) also facilitates FL researchers to combine prompt learning with personalized FL (PFL). Instead of restricting to the global consensus model, PFL (Tan et al., 2022; Kulkarni et al., 2020) allows tailored models for each individual client, systemically mitigating the data heterogeneity issues and enhancing the overall flexibility of the system. However, directly applying PFL techniques, such as local fine-tuning (Cheng et al., 2021) and personalizing specific layers (Arivazhagan et al., 2019), to prompt learning has demonstrated limited efficacy in training the prompt (Guo et al., 2023b) under extreme data heterogeneity. Consequently, some works tailor PFL techniques to prompt learning in CLIP-like VLMs. In this recently emerged field, Guo et al. (2023a) trains a global consensus prompt with personalized visual attention modules on locally memorized data. Li et al. (2024) leverages unbalanced Optimal Transport to align visual feature maps with personalized prompts. More details on prior works are presented in Appendix A.

Unfortunately, existing federated prompt learning approaches **habitually adhere to the paradigm of traditional FL/PFL techniques**, wherein the participating clients are generally only allowed to download a single globally aggregated model from the server. While justifiable for full-sized models, we argue that this paradigm is ill-suited for federated prompt learning, as the significantly reduced size of prompts substantially mitigates the potential communication overhead. To contextualize this disparity, consider that while a typical ResNet50 model contains approximately 25.6 million parameters, a learnable prompt with length 16 and dimension 512 comprises merely $16 \times 512 = 8,192$ parameters — a reduction of three orders of magnitude. Given this significant parameter reduction, restricting clients to downloading only a single globally aggregated model — as in traditional FL/PFL — unnecessarily limits federated prompt learning. **The lightweight nature of prompts provides a unique way for more flexible and effective personalization strategies.** Therefore, this motivates us to address the following pivotal question:

*How can we devise a personalized federated learning framework, tailored for prompt learning in CLIP-like VLMs, while fully exploiting the lightweight nature of the prompts?*

In light of these challenges and opportunities, we propose a novel framework: **P**ersonalized **Fed**erated **M**ixture **o**f **A**daptive **P**rompts (**`pFedMoAP`**). Tailored specifically for prompt learning in CLIP-like VLMs, our proposed framework aims to unleash the potential of the lightweight prompt

by allowing the clients to download **multiple pre-aggregated prompts** (see Fig. 1b) to acquire collective knowledge. In this manner, we enhance the generalization capabilities of the PFL system under extreme data heterogeneity, offering a more flexible and effective solution that fully leverages the lightweight nature of prompts in federated settings.

At its core, pFedMoAP personalizes the federated prompt learning problem through the lens of Mixture of Experts (MoE), treating all locally updated prompts as specialized experts. Benefiting from lifting the aforementioned ill-suited restriction, pFedMoAP facilitates the sharing of pre-aggregated prompts between clients (through the server). In addition, our proposed framework implements a novel client-specific, parameter-efficient, attention-based gating network that learns to generate enhanced text features for better alignment with local image data on each client. Through this locally trained gating network, the enhanced text features are generated from the adaptive local and non-local prompt experts via CLIP's text encoder. The local expert is trained exclusively with the client's data, while non-local experts, trained on other clients, are sparsely selected from a server-maintained pool based on $K$ nearest neighbors (KNN), and shared with the client without aggregation, fostering collective knowledge sharing across clients beyond the global aggregation.

We summarize our main contributions as follows:

- We pioneer a paradigm shift in federated prompt learning for VLMs by challenging the restriction on sharing pre-aggregated prompts between clients. Circumventing this ill-suited restriction for prompt learning unlocks more potential inherent in the lightweight prompts, facilitating more effective cross-client knowledge sharing. As such, we pave the way for more flexible federated prompt learning in VLMs.

- We propose pFedMoAP, a novel framework designed specifically for personalizing federated prompt learning in CLIP-like VLMs under data heterogeneity for image recognition tasks. pFedMoAP personalizes the prompt learning with a unique attention-based gating network. Thanks to its flexibility, this gating network has the potential to extend beyond federated learning for prompt-based VLMs.

- We validate the effectiveness of the proposed pFedMoAP through extensive experiments and ablation studies across 9 widely adopted datasets under various federated settings. The results verify the superiority of pFedMoAP over compared state-of-the-art methods.

## 2 PRELIMINARIES

### 2.1 PERSONALIZED FEDERATED LEARNING

Conventional federated learning (FL) aims to train a global consensus model for a federation of clients with similar data. As the most notable FL algorithm, FedAvg (McMahan et al., 2017) minimizes the global objective over $N$ clients defined as:

$$\min_{\boldsymbol{\theta}} F(\boldsymbol{\theta}) = \sum_{i=1}^{N} p_i F_i(\boldsymbol{\theta}), \qquad (1)$$

where $\boldsymbol{\theta}$ and $F_i(\cdot)$ represents the global model and the local objective of client $i$, respectively, and the weight $p_i$ is often set as $p_i = n_i/n$ with $n = \sum_i n_i$ where $n_i$ denotes the number of data samples on client $i$. In FedAvg, the local objective $F_i(\cdot)$ measures client $i$'s empirical loss, $\sum_{k=1}^{n_i} \mathcal{L}_i(\boldsymbol{\theta}|(\boldsymbol{x}_k, y_k))$, where $\mathcal{L}_i$ represents its loss and $\boldsymbol{x}_k$ is its $k$-th data sample with ground truth label $y_k$.

Compared to conventional FL, personalized FL (PFL) relaxes the number of models where each client $i$ is allowed to have its tailored model $\boldsymbol{\theta}_i$. The goal of PFL is, therefore, defined as:

$$\min_{\boldsymbol{\theta}_1, ..., \boldsymbol{\theta}_N} F(\boldsymbol{\theta}_1, ..., \boldsymbol{\theta}_N) = \min_{\boldsymbol{\theta}_1, ..., \boldsymbol{\theta}_N} \sum_{i=1}^{N} p_i F_i(\boldsymbol{\theta}_i). \qquad (2)$$

### 2.2 PROMPT LEARNING FOR CLIP-LIKE VLMS

To efficiently adapt pre-trained CLIP-like VLMs to downstream tasks, prompt learning methods (Zhou et al., 2022b;a) model a prompt's context words with learnable vectors. While zero-shot

transfer of CLIP leverages the fixed word embedding $\boldsymbol{W} = \{\boldsymbol{w}_1, ..., \boldsymbol{w}_l\}$ from hand-crafted prompt templates with $l$ context words (e.g. "a photo of a $<$class$>$."), prompt learning replace it with the learnable prompt $\boldsymbol{P} = \{\boldsymbol{p}_1, ..., \boldsymbol{p}_l\} \in \mathbb{R}^{l \times d}$ where $d$ is the dimension of the word embedding. The full prompt $\boldsymbol{P}^{(c)}$, consisting of the learnable prompt $\boldsymbol{P}$ with the embedding of a class label $c$, is then fed into the fixed text encoder, $g(\cdot)$. Together with the fixed image encoder $f(\cdot)$, the classification logit for class $c$ is then computed as a matching score between the image and text features. And the prediction probability of each class for image $\boldsymbol{x}$ is derived by taking a Softmax, controlled by temperature $\tau$, over the logits:

$$\text{logit}^{(c)} = \text{sim}(f(\boldsymbol{x}), g(\boldsymbol{P}^{(c)})), \tag{3}$$

$$p(\hat{y} = c|\boldsymbol{x}) = \frac{\exp(\text{logit}^{(c)}/\tau)}{\sum_{k=1}^{C} \exp(\text{logit}^{(k)}/\tau)}, \tag{4}$$

where $\text{sim}(\cdot)$ denotes a metric function (e.g. cosine similarity), $\hat{y}$ is the predicted label, and $C$ denotes the number of classes. With the probabilities, we can learn the prompt $\boldsymbol{P}$ by minimizing the cross-entropy loss.

## 2.3 MIXTURE OF EXPERTS

With a similar form to FL's, Mixture of Experts (MoE) aggregates the *output* of multiple experts or trained models, instead of the model parameters. For a general MoE system with $N$ experts, the output of MoE with input $x$ is defined as:

$$MoE(\boldsymbol{x}) = \sum_{i=1}^{N} G(\boldsymbol{x})_i \cdot E_i(\boldsymbol{x}), \tag{5}$$

where $E_i(\cdot)$ represents the $i$-th expert, and $G(\cdot)_i$ represents the weight assigned to the output of expert $i$ from a $N$ dimensional vector, based on its input. The mechanism for weight assignment is often controlled by a network called **gating network** (also known as a router). The implementation of the gating network varies (Clark et al., 2022; Hazimeh et al., 2021; Zhou et al., 2022c), but a simple yet effective one is implemented by taking the Softmax over top-$K$ ($K \leq N$) logits of a linear layer with the remaining $N - K$ weights set to zero (Shazeer et al., 2017). We propose a novel lightweight attention-based gating network devised for CLIP-like VLMs that extends beyond federated learning. Details are presented in Sec. 3.3.

## 3 PERSONALIZED FEDERATED MIXTURE OF ADAPTIVE PROMPTS

**Motivation.** While federated prompt learning for pre-trained CLIP-like VLMs offers an approach to efficiently adapt these models to downstream tasks under federated settings, such methods mostly lack personalization for the prompt to generalize with extreme data heterogeneity. Prior PFL methods devised for prompt learning habitually adhere to the restriction for the clients to download only a single globally aggregated model from the server, failing to leverage the lightweight nature of the prompt to the fullest. In addition, from an MoE perspective, allowing clients to download pre-aggregated prompts trained on other clients naturally provides a systemic solution to the dilemma where too many experts on the client incurs prohibitively high communication overhead while too few experts impairs the benefit of employing MoE. Consequently, there is a need for a tailored PFL approach for prompt learning in CLIP-like VLMs that offers the flexibility to share trained prompts from an MoE perspective.

**Overview.** The rest of this section presents the details of the `pFedMoAP` by gradually building upon PFL with prompt learning and mixture of adaptive prompts, and how the proposed attention-based gating network works in `pFedMoAP`, as well as potentials in extending it beyond federated learning.

### 3.1 PERSONALIZED FEDERATED PROMPT LEARNING WITH LOCAL PROMPT ONLY

In our study, we presume each client hosts a CLIP model with fixed image encoder $f(\cdot)$ and text encoder $g(\cdot)$ for image recognition tasks. We first suppose, in this subsection, that each client

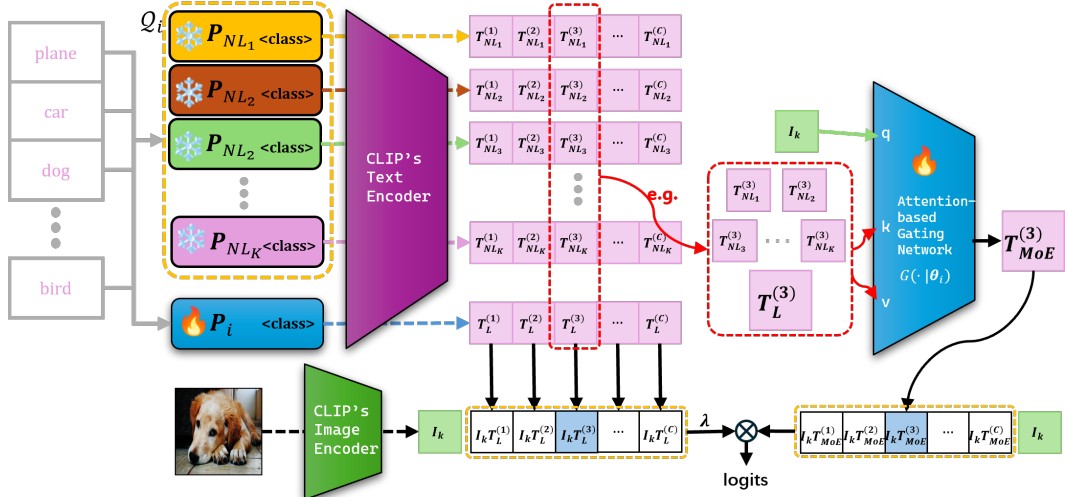

Figure 2: Workflow of `pFedMoAP` at client $i$. The client first computes the non-local text features using the non-local prompt experts. As training progresses, it then calculates the local text features. Taking class 3 as an example, both local and non-local text features are input into the attention-based gating network as both key and value, while image features serve as the query. This process generates enhanced text features. Matching socres are derived from two sources: local text features and MoE-enhanced text features. These scores are then combined through weighted averaging to produce the final logits.

personalizes the local prompt without the MoE. At federated round $t$, each client $i$ in the selected set of participating client $\mathcal{S}_t$ locally updates its prompt, $\boldsymbol{P}_i^t$, initialized with the global prompt from the last round, $\boldsymbol{P}_g^{t-1}$. The prompt is then updated through a gradient-based optimization, e.g. Stochastic Gradient Descent (SGD), over a cross-entropy loss for multiple local epochs.

After finishing the local update, the learned prompts $\boldsymbol{P}_i$ for all the clients in $\mathcal{S}_t$ are then aggregated by the server for a global prompt $\boldsymbol{P}_g^t$ in a `FedAvg` manner:

$$\boldsymbol{P}_g^t = \sum_{i \in \mathcal{S}_t} \frac{n_i}{\sum_{k \in \mathcal{S}_t} n_k} \boldsymbol{P}_i^t. \tag{6}$$

While $\boldsymbol{P}_g^t$ is used for the next round of training, for client $i$, $\boldsymbol{P}_i^t$ is stored and used for inference purposes, achieving a simple personalization for federated prompt learning while MoE is not present.

### 3.2 Personalized Federated Mixture of Adaptive Prompts workflow

Thanks to the lightweight nature of the prompt, clients are enabled to acquire collective knowledge beyond the globally aggregated prompt. In `pFedMoAP`, we facilitate, rather than discourage, each client to download $K$ pre-aggregated prompts trained on other clients as non-local client experts. These non-local experts are sparsely selected from a server-maintained pool of prompt experts, $\mathcal{P}$. The pool functions as a dynamic repository, refreshing at the conclusion of each federated round. It incorporates the newly acquired prompts from the clients who participated in the current training round, and overwrites their previous entries (if existing) in the pool, i.e:

$$\mathcal{P}_t = \mathcal{P}_{t-1} - \{\boldsymbol{P}_i^{t-1}\}_{i \in \mathcal{P}_{t-1} \cap \mathcal{S}_t} + \{\boldsymbol{P}_j^t\}_{j \in \mathcal{S}_t}. \tag{7}$$

At the beginning of the current round $t$, each client $i \in \mathcal{S}_t$ is first assigned $K$ non-local experts from the previous round's pool, $\mathcal{P}_{t-1}$. This assignment is based on a $K$ nearest neighbors (KNN) algorithm, which identifies the top-$K$ nearest experts from client $i$'s entry in the pool, in terms of $l_2$ distance[1]. the rationale behind the KNN-based expert assignment is that clients with similar locally

---

[1]The first-time participants with no entry in the pool conduct a standard local training of the global prompt, and upload its trained prompt to the pool.

trained prompts are more likely to share comparable data distributions. This similarity-based expert assignment aims to leverage knowledge from clients with potentially similar data characteristics, enhancing the personalization from closer collaborators in terms of local data distributions.

While such a server-side assignment of the non-local experts inherently simulates the *sparse* gating in MoE, a client-side gating network, $G_i(\cdot)$, enables dynamic *dense* incorporation of the output from all selected experts on a per-image basis (see Fig. 2).

Let $\mathcal{Q}_i = \{NL_j\}_{j=1}^K$ ($NL$ represents "*Non-Local*") be the set of clients assigned to client $i$, whose prompts, $\boldsymbol{P}_{NL_j}$, are served as non-local experts on client $i$. Before the local training of client $i$ initiates, text features for every non-local prompt are computed via the CLIP's fixed text encoder, $g(\cdot)$. Note that since the non-local prompt experts are kept fixed throughout the local training, these text features are only necessary to be computed once. Let $\boldsymbol{T}_{NL_j}^{(c)}$ be the text features for class $c$ generated by the $j$-th non-local prompt expert on client $i$ with class $c$'s embedding, $\boldsymbol{P}_{NL_j}^{(c)}$. At the beginning of the local training on client $i$, the text features, $\boldsymbol{T}_{NL}^{(c)}$, from every non-local expert for every class $c \in [C]$, formulated below, are fixed and ready to use for the entire local training process:

$$\forall c \in [C], \ \boldsymbol{T}_{NL}^{(c)} \triangleq \{\boldsymbol{T}_{NL_j}^{(c)} | \boldsymbol{T}_{NL_j}^{(c)} = g(\boldsymbol{P}_{NL_j}^{(c)}), \ \forall NL_j \in \mathcal{Q}_i\}. \tag{8}$$

In `pFedMoAP`, a novel attention-based gating network, $G(\cdot|\boldsymbol{\theta}_i)$ parameterized by $\boldsymbol{\theta}_i$, is proposed. Unlike the $G(\cdot)$ in Eq. (5), the proposed gating network functions to generate a **mixture of the outputs from the experts**, instead of the weights for the outputs in traditional MoE. The weights, however, are still internally computed as intermediate products through the attention mechanism. We discuss the gating network in details in Sec. 3.3. Let $\boldsymbol{I}_k = f(\boldsymbol{x}_k)$ be the computed image feature of the input image $\boldsymbol{x}_k$ through CLIP's fixed image encoder $f(\cdot)$. In `pFedMoAP`, for each class $c \in [C]$, what the gating network, $G(\cdot|\boldsymbol{\theta}_i)$, takes as input are three-fold: **1)** image feature, $\boldsymbol{I}_k$; **2)** text features from the locally updated prompt expert $\boldsymbol{T}_L^{(c)} = g(\boldsymbol{P}_i^{(c)})$ where $L$ represents "*Local*"; and **3)** text features from the all non-local experts, $\boldsymbol{T}_{NL}^{(c)}$ defined in Eq. (8). The gating network then outputs the enhanced text feature, $\boldsymbol{T}_{MoE}^{(c)}$, better aligned with the image feature $\boldsymbol{I}_k$ for more generalizability that incorporates both collective knowledge and personalization, i.e.,

$$\forall c \in [C], \ \boldsymbol{T}_{MoE}^{(c)} \triangleq G(\boldsymbol{I}_k, \boldsymbol{T}_L^{(c)}, \boldsymbol{T}_{NL}^{(c)}|\boldsymbol{\theta}_i). \tag{9}$$

Note that although the gating network could technically be parameterized in a class-specific manner, our implementation opts to use a shared set of parameters across all classes for parameter efficiency.

As the final step, `pFedMoAP` computes the logits based on the matching score between the image feature $\boldsymbol{I}_k$ and text features from two sources, namely $\boldsymbol{T}_{MoE}^{(c)}$ from the MoE, and $\boldsymbol{T}_L^{(c)}$ from the local expert. Although $\boldsymbol{T}_{MoE}^{(c)}$ carries the global collective knowledge, we further address $\boldsymbol{T}_L^{(c)}$ with weight $\lambda$ as the local prompt $\boldsymbol{P}_i$ is the only learnable expert on the client, i.e.,

$$\forall c \in [C], \ \text{logit}^{(c)} = \text{sim}(\boldsymbol{I}_k, \boldsymbol{T}_{MoE}^{(c)}) + \lambda \cdot \text{sim}(\boldsymbol{I}_k, \boldsymbol{T}_L^{(c)}). \tag{10}$$

With the classification logits, the local prompt $\boldsymbol{P}_i$ and the parameter $\boldsymbol{\theta}_i$ in the gating network are updated through optimizing over the cross-entropy loss based on the prediction probability computed as in Eq. (4). After the local training, client $i$ upload $\boldsymbol{P}_i$ to the server while maintaining the gating network locally.

### 3.3 Attention-based Gating Network: in `pFedMoAP` and Beyond

To generate enhanced text features that better aligns with the image feature $\boldsymbol{I}_k$, the proposed gating network employs a multi-head attention (MHA) layer (Vaswani, 2017) with the query being the $\boldsymbol{I}_k$, while the key and the value are both text features $\boldsymbol{T}_L^{(c)}, \boldsymbol{T}_{NL_1}^{(c)}, \boldsymbol{T}_{NL_2}^{(c)}, ..., \boldsymbol{T}_{NL_K}^{(c)}$. However, since the dimension of these features are usually large, resulting in too many parameters in the MHA layer (e.g. for CLIP with a ResNet50 backbone, $d_{\text{feature}} = 1024$, MHA layer will have 4.2M parameters), we force each feature to first go through a pooling layer to reduce the dimension to a fixed $d_{\text{gating}} = 128$ (MHA layer will only have 66.0K parameters). Therefore, with $Q = \text{Pooling}(\boldsymbol{I}_k), K = V = \text{Pooling}(\boldsymbol{T}_L^{(c)}, \boldsymbol{T}_{NL}^{(c)})$ and $h$ being then number of heads, we have:

$$\boldsymbol{T}_{MoE}^{(c)} = G(\boldsymbol{I}_k, \boldsymbol{T}_L^{(c)}, \boldsymbol{T}_{NL}^{(c)}|\boldsymbol{\theta}_i) = \text{MHA}(Q, K, V) = \text{Concat}(\text{head}_1, ..., \text{head}_h)W^O, \tag{11}$$

Table 1: Few-shot performance on CLIP datasets under pathological non-IID setting.

| | Flowers102 | OxfordPets | Food101 | Caltech101 | DTD |
|---|---|---|---|---|---|
| ZS-CLIP (Radford et al., 2021) | 62.17±0.12 | 84.47±0.01 | 75.27±0.05 | 85.14±0.24 | 40.21±0.12 |
| CoOp (Zhou et al., 2022b) | 70.14±0.76 | 83.21±1.30 | 70.43±2.42 | 87.37±0.44 | 44.23±0.63 |
| PromptFL (Guo et al., 2023b) | 72.80±1.14 | 90.79±0.61 | 77.31±1.64 | 89.70±1.99 | 54.11±0.22 |
| PromptFL+FT (Cheng et al., 2021) | 72.31±0.91 | 91.23±0.50 | 77.16±1.56 | 89.70±0.25 | 53.74±1.36 |
| Prompt+FedPer (Arivazhagan et al., 2019) | 72.11±1.35 | 89.50±1.62 | 71.29±1.87 | 86.72±1.45 | 50.23±0.82 |
| Prompt+FedProx (Li et al., 2020b) | 66.40±0.29 | 89.24±0.41 | 76.24±1.94 | 89.41±0.55 | 44.26±1.11 |
| Prompt+FedAMP (Huang et al., 2021) | 69.10±0.13 | 80.21±0.44 | 74.48±1.71 | 87.31±1.60 | 47.16±0.92 |
| pFedPrompt (Guo et al., 2023a) | 86.46±0.15 | 91.84±0.41 | 92.26±1.34 | 96.54±1.31 | 77.14±0.09 |
| FedOTP (Li et al., 2024) | 96.23±0.44 | 98.82±0.11 | 92.73±0.15 | 97.02±0.36 | 87.64±0.70 |
| pFedMoAP ($\lambda$=0.0) | 97.61±0.11 | 94.83±0.65 | 86.71±0.15 | 95.71±0.37 | 85.64±0.34 |
| pFedMoAP ($\lambda$=0.5) | 98.41±0.04 | 99.06±0.09 | 93.39±0.09 | 97.95±0.07 | 89.13±0.54 |

where $\text{head}_q = \text{Attention}(QW_q^Q, KW_q^K, VW_q^V)$ and $W^O, W_q^Q, W_q^K, W_q^V$ are standard MHA parameters in $\boldsymbol{\theta}_i$. As $\boldsymbol{T}_{MoE}^{(c)} \in \mathbb{R}^{d_{\text{gating}}}$, computing $\text{sim}(\boldsymbol{I}_k, \boldsymbol{T}_{MoE}^{(c)})$ requires pooling on $\boldsymbol{I}_k$ as well.

While a linear projection-based gating network, $G_{\text{proj}}(\cdot)$, in a typical MoE is also able to achieve enhanced text features for CLIP (e.g. by assigning weights $G_{\text{proj}}(\boldsymbol{x}_k) \in \mathbb{R}^{K+1}$ for the experts' output based on the image input (Shazeer et al., 2017)), the proposed attention-based gating network is more favorable for the following FL-agnostic reasons. **1)** Since the experts are constantly adapting during the training process, directly learning the assigned weights for the experts, as in projection-based gating, will be less robust than learning the relationship between the image feature and the text features generated by the experts, as in attention-based gating. **2)** The output of the proposed attention-based gating is the enhanced text features, which makes the MHA layer serve as a lightweight linear probing with a much larger search space, that functions beyond assigning $K + 1$ weights to aggregate the experts' output as a linear combination in a $K + 1$-dimensional search space. Even with a redesigned projection layer to directly produce the enhanced text features, the parameter count of such a projection-based gating network would substantially exceed that of the attention-based gating network. **3)** A projection-based gating network fails to leverage the pre-trained CLIP's powerful capability to align text features with image features, while the attention mechanism naturally harnesses it through the scaled dot-product operation. **4)** Even if we feed both image and text features from multiple experts into a projection-based gating network, the order of these input would affect the final output in an uncontrollable way. And the number of experts has to be fixed for a projection-based gating network. In contrast, the proposed attention-based gating network is order-agnostic with a flexible number of experts it engages with. In pFedMoAP, while we fix the number of non-local experts on each client to be $K$, in practice, $K$ can be client-specific based on the client's memory and communication bandwidth.

As such, our design of the attention-based gating network extends beyond federated learning. This versatile network could be adapted to general prompt learning for VLMs where multiple prompts are trained, such as multi-task learning or domain adaptation. We anticipate that our work will inspire future research exploring the broader applications of mixture of adaptive prompts for VLMs. The detailed pFedMoAP algorithm can be found in Algorithm 1.

# 4 EXPERIMENTS

## 4.1 EXPERIMENTAL SETUP

**Dataset and data heterogeneity.** We evaluate the efficacy of the proposed pFedMoAP with 9 public benchmark datasets under various federated settings to simulate different types of data heterogeneity. Following previous research (Guo et al., 2023b), to evaluate pFedMoAP under label heterogeneity, we adopt 5 representative visual classification datasets used to evaluate CLIP (Radford et al., 2021), namely OxfordPets Parkhi et al. (2012), Flowers102 Nilsback & Zisserman (2008), DTD Cimpoi et al. (2014), Caltech101 Fei-Fei (2004), Food101 Bossard et al. (2014). We refer to these datasets collectively as CLIP datasets. On these datasets, we test pFedMoAP's few-shot performance under label heterogeneity by employing a pathological non-IID setting, where the classes are evenly distributed to the clients with no overlapping classes between any two clients. In addi-

tion, we use CIFAR10 and CIFAR100 dataset (Gong et al., 2012) and a Dirichlet distribution with $\text{Dir}(\alpha = 0.5)$ to simulate the label shift (Hsu et al., 2019). The Dirichlet distribution with $\alpha = 0$ represents an extreme case where each client has only one class and larger $\alpha$ simulates a stratified and even split of labels. $\text{Dir}(\alpha = 0.5)$ is on the more heterogeneous side. For feature heterogeneity, we adopt DomainNet dataset (Peng et al., 2019) and Office-Caltech10 dataset (Gong et al., 2012), with 6 and 4 inherent domains, respectively. To add on extra heterogeneity, for DomainNet and Office-Caltech10, we split each domain with $\text{Dir}(\alpha = 0.3)$ for extra label heterogeneity, mimicking real-world FL settings with both label and feature shifts.

**Compared baselines.** We compare our approach with three categories of baselines. The first category is local methods, where the clients do not participate in a FL setting. This category includes zero-shot CLIP (`ZS-CLIP`) (Radford et al., 2021) employing prompt templates: "a photo of a <class>", and `CoOp` (Zhou et al., 2022b) with learnable prompt vectors trained locally on each client. The second category comprises existing federated prompt learning methods, which include `PromptFL` (Guo et al., 2023b), which learns a unified prompt across clients, `pFedPrompt` (Guo et al., 2023a), which learns a shared prompt with personalized visual attention modules on locally memorized data, and `FedOTP` (Li et al., 2024), which leverages unbalanced Optimal Transport to align visual feature maps with personalized prompts. The third category consists of adapted methods from traditional FL/PFL techniques on `PromptFL`, namely `FT` (Cheng et al., 2021), `FedProx` Li et al. (2020b), `FedPer` (Arivazhagan et al., 2019), and `FedAMP` (Huang et al., 2021).

**Implementation details.** **1)** CLIP datasets. Each dataset in CLIP datasets is partitioned into $N = 10$ clients, each with a disjoint set of classes evenly and randomly assigned to the clients. The training proceeds for $T = 10$ rounds with $r = 100\%$ participation rate. The CLIP uses a ResNet50 backbone. **2)** CIFAR10 & CIFAR100. $N = 100$ clients results from $\text{Dir}(\alpha =$

Table 2: Results on CIFAR10 and CIFAR100 with label shift with ($\text{Dir}(\alpha = 0.5)$) partition into 100 clients

|  | CIFAR10 | CIFAR10 |
|---|---|---|
| `ZS CLIP` (Radford et al., 2021) | 53.46±0.21 | 32.68±0.00 |
| `CoOp` (Zhou et al., 2022b) | 80.84±0.39 | 48.74±0.17 |
| `PromptFL` (Guo et al., 2023b) | 73.29±0.37 | 45.00±0.62 |
| `Prompt+FedProx` (Li et al., 2020b) | 73.32±0.34 | 45.63±0.75 |
| `pFedMoAP` | 83.46±0.53 | 53.42±0.22 |

$0.5)$ partition with $T = 120$ rounds with $r = 10\%$. CLIP also uses a ResNet50 backbone. **3)** DomainNet & Office-Caltech10. Each domain of these two datasets is partitioned to 5 clients with $\text{Dir}(\alpha = 0.3)$, resulting in $N = 30$ for DomainNet and $N = 20$ for Office-Caltech10. $T = 25$ for both datasets, while $r = 25\%$ for DomainNet and $r = 50\%$ for Office-Caltech10. The CLIP uses a ViT-b-16 backbone. **4)** Training specs. For all methods, we use SGD as the optimizer with a learning rate of 0.002 and 5 local epochs (except `CoOp` is locally trained for 25 epochs without FL). For `pFedMoAP`, we use SGD with a learning rate of 0.01 to train the $h = 8$-head gating network and default $\lambda = 0.5$ in Eq. (10). A summary of the dataset setup can be found at Appendix C.1.

## 4.2 PERFORMANCE EVALUATION

We compute as the metric the average accuracy over every client's private test set, drawn from the same distribution as its training set, and report the mean and standard deviation of all methods over three runs with different seeds.

**Evaluation on Label Shifts.** To assess the performance of our method in handling label shifts, we conducted experiments on the CLIP datasets for few-shot training and CIFAR datasets for standard training. Tab. 1 shows the results on the CLIP datasets under a pathological non-IID setting, while Tab. 2 presents the results on CIFAR10 and CIFAR100 with $\text{Dir}(\alpha = 0.5)$. On the CLIP datasets (Tab. 1), our proposed method `pFedMoAP` consistently outperforms other baselines across all five datasets, with significant margins over most methods, which proves the superiority of `pFedMoAP`. For instance, on the Flowers102 dataset, `pFedMoAP` achieves an accuracy of 98.41% with $\lambda = 0.5$, respectively 2.18% and 11.95% higher than `FedOTP` and `pFedPrompt`, the top 2 performant federated prompt learning methods, both recently proposed to achieve personalization through intermediate visual feature maps. Similar improvements can be observed across other datasets. Notably, on DTD dataset where zero-shot struggles to excel, `pFedMoAP` has little performance drop compared to personalized methods such as `pFedPrompt` and `PromptFL+FedPer`, showing `pFedMoAP`'s

Table 3: Results on DomainNet with feature shift and label shift with Dir($\alpha = 0.3$) partition into 5 clients/domain

|  | Clipart | Infograph | Painting | Quickdraw | Real | Sketch | Average |
|---|---|---|---|---|---|---|---|
| ZS CLIP | 9.18±0.62 | 10.03±0.16 | 9.93±0.51 | 10.25±0.40 | 9.90±1.30 | 9.54±1.13 | 9.81±0.30 |
| CoOp | 43.84±3.51 | 45.72±0.85 | 29.94±0.46 | 36.83±1.17 | 31.64±0.49 | 33.97±0.78 | 36.99±0.79 |
| PromptFL | 27.63±16.41 | 27.69±18.07 | 21.62±8.34 | 23.45±13.49 | 20.62±11.03 | 25.90±8.10 | 24.48±12.52 |
| Prompt+FedProx | 22.23±15.42 | 21.75±17.00 | 18.58±8.15 | 19.40±12.59 | 17.17±10.25 | 22.49±8.44 | 20.27±11.83 |
| pFedMoAP | 47.49±0.64 | 46.73±0.71 | 32.74±0.84 | 37.16±0.34 | 31.02±0.59 | 37.67±0.72 | 38.80±0.11 |

Table 4: Results on Office-Caltech10 with feature shift and label shift with Dir($\alpha = 0.3$) partition into 5 clients/domain

|  | Amazon | Caltech | DSLR | Webcam | Average |
|---|---|---|---|---|---|
| ZS-CLIP (Radford et al., 2021) | 9.83±1.63 | 10.67±0.89 | 10.89±1.40 | 6.20±3.84 | 9.40±0.77 |
| CoOp (Zhou et al., 2022b) | 30.29±3.64 | 35.88±1.30 | 29.89±5.15 | 33.43±2.25 | 32.37±1.81 |
| PromptFL (Guo et al., 2023b) | 21.08±9.60 | 23.72±12.21 | 22.94±7.96 | 25.88±7.72 | 23.41±9.06 |
| Prompt+FedProx (Li et al., 2020b) | 18.64±8.58 | 19.56±11.59 | 20.89±7.38 | 22.96±7.56 | 20.51±8.48 |
| pFedMoAP | 35.47±1.37 | 37.45±1.33 | 45.11±3.14 | 35.22±1.04 | 38.31±1.21 |

remarkable ability to adapt to diverse scenarios under few-shot settings. In addition, `pFedMoAP` with $\lambda = 0.5$ also beats $\lambda = 0.0$ over all datasets, demonstrating the effectiveness of addressing the local prompt together with MoE. The results on CIFAR datasets (Tab. 2) further corroborate the efficacy of our method in handling extreme data heterogeneity with Dir($\alpha = 0.5$) over 100 clients. In this challenging scenario, `pFedMoAP` still consistently outperforms the compared baselines, reflecting the potency to handle label shifts by introducing non-local experts to the clients.

**Evaluation on Feature & Label Shifts.** To evaluate the performance of our method in scenarios closer to real-world FL applications, we introduce feature shift on top of label shift via DomainNet and Office-Caltech10 where each domain is partitioned into 5 clients with Dir($\alpha = 0.3$), resulting in 30 and 20 total clients in DomainNet and Office-Caltech10, respectively. The results of these are presented in Tabs. 3 and 4, respectively. Under two types of heterogeneity, traditional federated learning methods struggle to benefit the clients. On average, the `FedAvg` style method `PromptFL` exhibits worse performance than the local training method `CoOp` by 12.51% and 8.96% respectively on DomainNet and Office-Caltech10. However, `pFedMoAP` remains better than local training and FL with up to 5.94% accuracy boost under both types of heterogeneity. This validates the effectiveness and robustness of `pFedMoAP` in scenarios closer to real-world federated settings.

## 4.3 ABLATION STUDY

**Impact of number of shot.** Following prior works (Li et al., 2024; Guo et al., 2023a), we investigate the impact of shots in the few-shot learning for `pFedMoAP` and other FL techniques implemented to prompt learning, namely `PromptFL` and `PromptFL+FedProx`, across the CLIP datasets. The number of shots varies from [1, 2, 4, 8, and 16]. As shown in Fig. 3, `pFedMoAP` consistently outperforms the other two methods across all datasets and shot numbers, demonstrating its superior effectiveness and robustness. While `pFedMoAP` outperforms the compared methods from only 1 shot, the gap between `pFedMoAP` and other methods widens as the number of shots increases. This demonstrates that `pFedMoAP` is particularly adept at leveraging additional training examples to enhance its performance, due to the introduction of MoE and the gating network. The trend of the curves also just that `pFedMoAP` has a more robust growth of performance when the number of shots increases, where the performances of the compared methods sometimes drop.

**Impact of $\lambda$, the coefficient for logit computed from local prompt.** In `pFedMoAP`, $\lambda$ balances the contribution between the logits generated from the local prompt and from MoE. Fig. 4 displays the results for $\lambda$ values ranging from 0.0 to 5.0 across the CLIP datasets. The trends reveal that although the optimal $\lambda$ value varies across datasets, a marginal address on the logits from the local prompt (e.g. $\lambda = 0.5$ or 1) can already reach near-best performance. The flat plateau also indicates that the balance between the logits generated from the local prompt and from MoE is easy to reach.

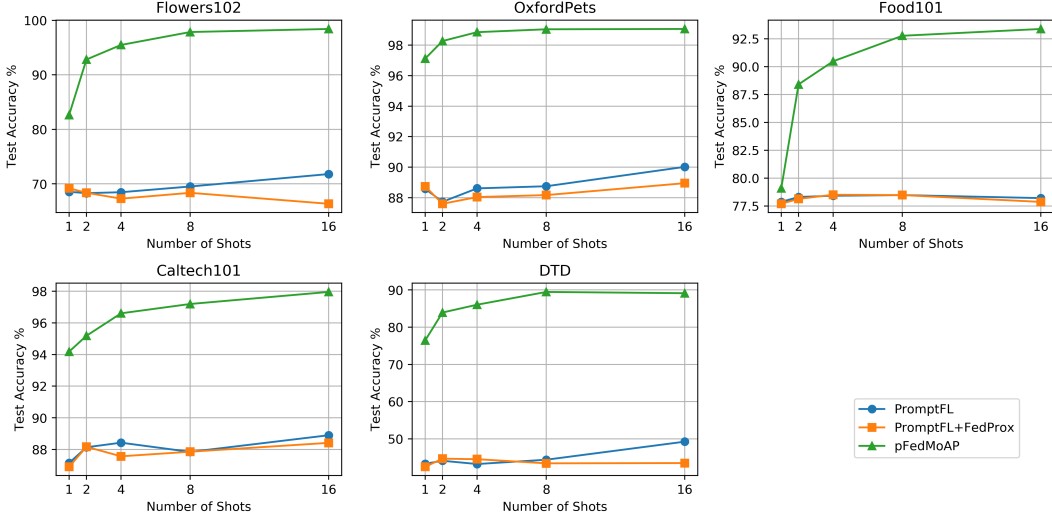

Figure 3: Ablation study on the number of shots.

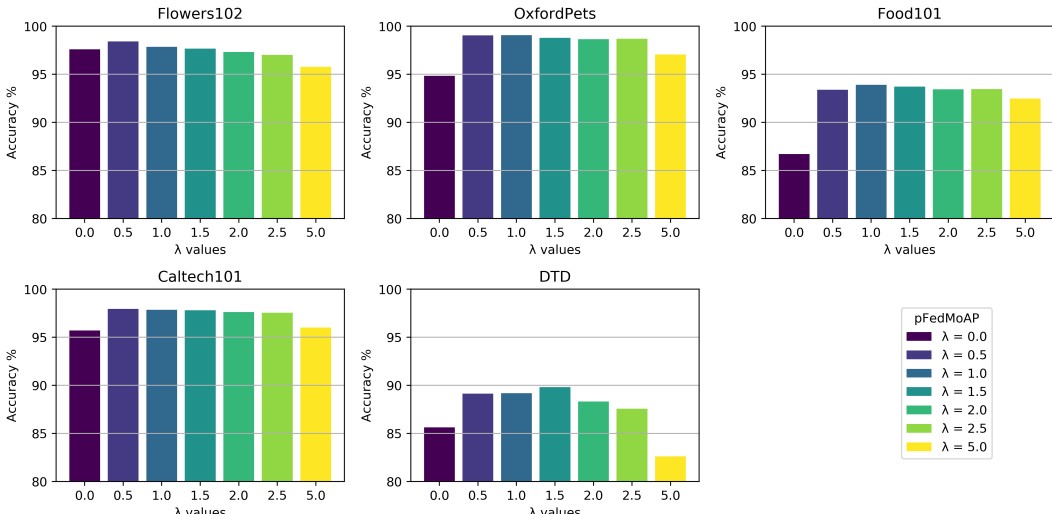

Figure 4: Ablation study on the coefficient for the logits from local prompt, $\lambda$.

In addition, Fig. 4 also demonstrate a disinclination of too little or too much address on the local prompt-generated logits. We attribute this to the fact that only the local prompt is being updated on a client among all the experts, while immoderately addressing it ignores the benefit from MoE.

## 5 CONCLUSION

In this paper, we introduced `pFedMoAP`, a novel framework for personalized federated prompt learning in CLIP-like VLMs. Our approach leverages the lightweight nature of prompts to enable efficient cross-client knowledge sharing through a Mixture of Experts paradigm. By implementing a client-specific, attention-based gating network, `pFedMoAP` dynamically incorporates both local and non-local expert knowledge, striking a balance between personalization and collaborative learning. Extensive experiments across various datasets and federated settings demonstrate the superior performance of `pFedMoAP` compared to state-of-the-art alternatives, particularly in handling extreme data heterogeneity with label and/or feature shift. As federated learning continues to evolve, `pFedMoAP` represents a significant step forward in personalizing prompt learning for VLMs, opening new avenues for efficient and effective collaborative learning in privacy-preserving settings.

ACKNOWLEDGMENTS

This work was supported in part by a National Institutes of Health (NIH)/National Cancer Institute (NCI) grant (1R01CA218405), a NIH Other Transaction research contract 1OT2OD037972-01, the grant 1R01EB032896 (and a Supplement grant 3R01EB03 2896-03S1) as part of the National Science Foundation (NSF)/NIH Smart Health and Biomedical Research in the Era of Artificial Intelligence and Advanced Data Science Program, a NSF grant (CICI: SIVD: #2115082), an Amazon Machine Learning Research Award, and the University of Pittsburgh Momentum Funds (a scaling grant) for the Pittsburgh Center for AI Innovation in Medical Imaging. This work used Bridges-2 at Pittsburgh Supercomputing Center through allocation [MED200006] from the Advanced Cyberinfrastructure Coordination Ecosystem: Services & Support (ACCESS) program, which is supported by NSF grants #2138259, #2138286, #2138307, #2137603, and #2138296. The views and conclusions contained in this document are those of the authors and should not be interpreted as representing official policies, either expressed or implied, of the NIH or NSF.

**Ethics Statement.** The research presented in this paper adheres to the ICLR Code of Ethics. Our study does not involve human or animal subjects. Every dataset used in our study is publicly available. Our method does not address discrimination/bias/fairness concerns.

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

# APPENDIX

## A  RELATED WORK

**Personalized federated learning.**  The mediocre performance of conventional federated learning (FL) (McMahan et al., 2017) over heterogeneous data calls for a more customized solution. Personalized federated learning (PFL) (Tan et al., 2022; Kulkarni et al., 2020), allowing tailored models for each client instead of a consensus model, systemically mitigates the data heterogeneity issue. Various strategies have been proposed to achieve personalization, including local fine-tuning (Cheng et al., 2021), different optimization techniques such as attentive message passing and regularization (Huang et al., 2021; Li et al., 2021a), and different personalized layers including feature extractor, classification head, attention, and batch normalization layers (Liang et al., 2020; Arivazhagan et al., 2019; Sun et al., 2023; Li et al., 2023; 2021b). Some works use separate models to learn global and local knowledge, respectively (Cai et al., 2024; Deng et al., 2020). Some recent PFL works also switch gear to model heterogeneity (Xie et al., 2024; Yi et al., 2024a). While promising in addressing data heterogeneity, these methods primarily focus on traditional machine learning models and do not leverage the capabilities of large pre-trained vision-language models.

**Federated prompt learning for pre-trained models.**  Prompt learning for pre-trained models combines models' generalization capabilities and prompt learning's flexibility in adapting these models to downstream tasks. As a fundamental work for this combination, `CoOp` (Zhou et al., 2022b) models a prompt's context words with learnable vectors in CLIP. Such a combination has quickly drawn the attention of the FL community. For instance, `PromptFL` (Guo et al., 2023b) does a `FedAvg` (McMahan et al., 2017) style global aggregation over the locally updated prompts. Building upon this, `FedPR` (Feng et al., 2023) learns federated visual prompts for MRI reconstruction, while `Fed-DPT` (Su et al., 2022) applies both visual and textual prompt tuning to facilitate domain adaptation over decentralized data. `pFedprompt` (Guo et al., 2023a) integrating global consensus prompt and local attention over stored few-shot data for each client. `FedOTP` (Li et al., 2024) and `FedPGP` (Cui et al., 2024) employ Optimal Transport and prompt-wise contrastive loss, respectively, between global and local prompts, capturing diverse category traits on a per-client basis. Pan et al. (2024) presents a theoretical analysis framework for prompt-based federated learning of vision-language models. However, these approaches habitually adhere to the paradigm where locally learned prompts are not allowed for inter-client sharing before aggregation. While sharing full models between clients would be impractical due to the massive communication costs $- O(N^2 \cdot M)$ per round when $N$ models of size $M$ are distributed to $N$ clients (Luo & Wu, 2022; Luo et al., 2023) – sharing prompts is a different scenario. Since prompts are orders of magnitude smaller than full models (as shown in Fig. 1), the communication overhead becomes much more affordable. This makes it feasible for the inter-client sharing of pre-aggregated prompts under federated settings.

**Federated learning with Mixture of Experts.**  From the proprietary GPT-4 (Achiam et al., 2023) to the open-sourced Mixtral of Experts $8\times7B$ (Jiang et al., 2024), Qwen1.5-MoE (Bai et al., 2023), Mixture of Experts (MoE) (Zhou et al., 2022c; Masoudnia & Ebrahimpour, 2014) is prevailing long after its initial proposal thanks to the recent heat in Large Language Models (LLMs). Its application in FL, however, dates back before LLMs. `FedMix` (Reisser et al., 2021) and `FedJETs` (Dun et al., 2023) allow each client to construct an MoE with a shared gating network, selecting specific non-local models more adaptive to the client's local data for ensembling. `PFL-MoE`(Guo et al., 2021) and `pFedMoE` (Yi et al., 2024b) incorporate the global aggregated model as the global expert and its locally fine-tuned model as a local expert to each client, achieving a personalized two-expert MoE. However, under federated settings, with too many experts on the client comes prohibitively high communication overhead (Reisser et al., 2021; Dun et al., 2023) while too few experts impairs the benefit of employing MoE.

In contrast with the aforementioned methods, our proposed `pFedMoAP`, is a devised PFL method for prompt learning in CLIP-like VLMs. It sidesteps the impractical restriction on sharing the prompts without aggregation, while allowing a many-expert (e.g. 10 experts) MoE scenario for each client with negligible communication overhead, achieving efficient and effective personalization for the federated prompt learning.

## B  pFedMoAP: The algorithm

We provide the overall algorithm of pFedMoAP below.

---

**Algorithm 1** pFedMoAP

---

**Input:** $N$ clients, learning rates $\eta_1, \eta_2$, number of rounds $T$, logit coefficient $\lambda$, CLIP image/text encoder $f(\cdot), g(\cdot)$, datasets $\{D_i\}_{i \in [N]}$

**Output:** Personalized prompts $\boldsymbol{P}_1, \boldsymbol{P}_2, ..., \boldsymbol{P}_N$, gating network weights $\boldsymbol{\theta}_1, \boldsymbol{\theta}_2, ..., \boldsymbol{\theta}_N$.

**ServerExecute:**
1:  Server initialize $\boldsymbol{P}_g^0$ and the pool of prompt experts $\mathcal{P}_0$ as an empty set
2:  Clients intialize $\boldsymbol{\theta}_1, \boldsymbol{\theta}_2, ..., \boldsymbol{\theta}_N$.
3:  **for** $t \leftarrow 1, 2, ..., T$ **do**
4:      Select a subset of $|\mathcal{S}_t|$ clients, $\mathcal{S}_t$
5:      **for** $i \in \mathcal{S}_t$ **in parallel do**
6:          **if** Client $i$ does not have an entry in the server-maintained pool, $\mathcal{P}_t$ **then**
7:              $\boldsymbol{P}_i^t =$ClientUpdate($\boldsymbol{P}_g^{t-1}$, standard=True)
8:          **else**
9:              Compute $\mathcal{Q}_i$ by $K$ nearest neighbor, given $\boldsymbol{P}_i^t = \mathcal{P}_{t-1}[i]$.
10:             $\boldsymbol{P}_{NL} = \{\mathcal{P}_{t-1}[j]\}_{j \in \mathcal{Q}_i}$ // prompt in the pool from selected group of clients
11:             $\boldsymbol{P}_i^t =$ClientUpdate($\boldsymbol{P}_g^{t-1}$, $\boldsymbol{P}_{NL}$) // downloaded as non-local experts
12:             $\mathcal{P}_t[i] = \boldsymbol{P}_i^t$ // cache to pool
13:         **end if**
14:         $\boldsymbol{P}_g^t = \sum_{i \in \mathcal{S}_t} p_i \boldsymbol{P}_i^t$
15:     **end for**
16: **end for**
17: **return** $\boldsymbol{P}_1^T, \boldsymbol{P}_2^T, ..., \boldsymbol{P}_N^T$ and $\boldsymbol{\theta}_1, \boldsymbol{\theta}_2, ..., \boldsymbol{\theta}_N$

**ClientUpdate($\boldsymbol{P}_g^{t-1}$, $\boldsymbol{P}_{NL}$=None, standard=False):**
1:  $\boldsymbol{P}_i^t \leftarrow \boldsymbol{P}_g$
2:  **if** standard **then**
3:      client does a standard fine-tuning
4:  **else**
5:      **for** $(\boldsymbol{x}_k, y_k) \in D_i$ **do**
6:          $\boldsymbol{T}_L = g(\boldsymbol{P}_i^t)$
7:          $\boldsymbol{T}_{NL} = g(\boldsymbol{P}_{NL})$
8:          $\boldsymbol{I}_k = f(\boldsymbol{x}_k)$
9:          $\boldsymbol{T}_{MoE} = G(\boldsymbol{I}_k, \boldsymbol{T}_L, \boldsymbol{T}_{NL} | \boldsymbol{\theta}_i)$
10:         logit $= \text{sim}(\boldsymbol{I}_k, \boldsymbol{T}_{MoE}) + \lambda \cdot \text{sim}(\boldsymbol{I}_k, \boldsymbol{T}_L)$
11:         $p(\hat{y} = c | \boldsymbol{x}_k) = \text{Softmax}(\text{logit}, \tau)$ // $\tau$ is the temperature
12:         $\mathcal{L}_{ce} = -\sum_c y_k^{(c)} p(\hat{y} = c | \boldsymbol{x}_k)$
13:     **end for**
14: **end if**
15: **return** $\boldsymbol{P}_i^t$

---

## C  Additional experiments

### C.1  Details of dataset setup

We follow the most recent prompt-based FL for VLMs works' standard (Li et al., 2024; Cui et al., 2024) and use the 9 datasets they have used. Tab. 5 lists the details of these datasets and the partitioning method used in our experiments for each dataset.

Table 5: The statistics of each dataset and the partitioning details used in our experiments

| Dataset | Training set size | Test set size | #Classes | #Clients | Sample Rate | Heterogeneity |
|---|---|---|---|---|---|---|
| Flowers102 | 4,093 | 2,463 | 102 | 10 | 100% | Pathological |
| OxfordPets | 2,944 | 3,669 | 37 | 10 | 100% | Pathological |
| Food101 | 50,500 | 30,300 | 101 | 10 | 100% | Pathological |
| Caltech101 | 4,128 | 2,465 | 100 | 10 | 100% | Pathological |
| DTD | 2,820 | 1,692 | 47 | 10 | 100% | Pathological |
| Office-Caltech10 | 2,025 | 508 | 10 | 20 | 50% | Dir(0.3) |
| DomainNet | 18,278 | 4,573 | 10 | 30 | 25% | Dir(0.3) |
| CIFAR10 | 50,000 | 10,000 | 10 | 100 | 10% | Dir(0.5) |
| CIFAR100 | 50,000 | 10,000 | 100 | 100 | 10% | Dir(0.5) |

## C.2 pFedMoAP UNDER DIFFERENTIAL PRIVACY

In pFedMoAP, clients remain unaware of the origin of non-local prompts, including which client they come from or the training round they were last updated in. Besides, the non-local prompts are downloaded after a KNN-based sparse selection process on the server. These factors collectively make it extremely difficult for a client to infer the gradient associated with a prompt or the data it has been trained with, which largely mitigates the privacy risk.

Table 6: Performance under $(\epsilon, \delta)$-differential privacy on CLIP datasets under pathological non-IID setting.

| | Flowers102 | OxfordPets | Food101 | Caltech101 | DTD |
|---|---|---|---|---|---|
| **Without differential privacy (from Tab. 1)** | | | | | |
| PromptFL (Guo et al., 2023b) | 72.80±1.14 | 90.79±0.61 | 77.31±1.64 | 89.70±1.99 | 54.11±0.22 |
| PromptFL+FedProx (Li et al., 2020b) | 66.40±0.29 | 89.24±0.41 | 76.24±1.94 | 89.41±0.55 | 44.26±1.11 |
| pFedMoAP(ours) | 98.41±0.04 | 99.06±0.09 | 93.39±0.09 | 97.95±0.07 | 89.13±0.54 |
| **With differential privacy ($\epsilon = 50$)** | | | | | |
| PromptFL (Guo et al., 2023b) | 67.07±0.60 | 88.05±0.32 | 77.41±0.60 | 84.83±0.42 | 38.39±1.25 |
| PromptFL+FedProx (Li et al., 2020b) | 66.22±0.63 | 87.78±0.61 | 77.27±0.59 | 84.68±0.64 | 39.43±1.11 |
| pFedMoAP(ours) | 98.34±0.06 | 99.08±0.02 | 93.36±0.04 | 97.90±0.08 | 89.99±0.49 |
| **With differential privacy ($\epsilon = 25$)** | | | | | |
| PromptFL (Guo et al., 2023b) | 64.25±1.10 | 86.26±1.07 | 76.84±0.66 | 85.00±1.59 | 38.19±0.66 |
| PromptFL+FedProx (Li et al., 2020b) | 62.87±0.99 | 86.82±0.47 | 76.21±0.64 | 84.51±1.52 | 37.82±0.52 |
| pFedMoAP(ours) | 98.36±0.12 | 99.02±0.04 | 93.41±0.13 | 97.99±0.06 | 89.11±0.28 |

Additionally, we evaluate how $(\epsilon, \delta)$-differential privacy (DP) (Dwork, 2006) affects the performance of pFedMoAP on CLIP datasets and compare its performance with PromptFL and PromptFL+FedProx. This is achieved by clipping the gradients of the prompts and adding noise to the uploaded prompts. We set the failure probability $\delta$ to be $0.05$ and report the accuracy with the privacy budget $\epsilon$ being set to $50$ and $25$. The results in Tab. 6 demonstrate that while the accuracy of PromptFL and PromptFL+FedProx significantly drop with privacy guarantees, DP's impact in accuracy performance to pFedMoAP remains minimal for different privacy budgets. This is because the gating network in pFedMoAP is maintained locally for the entirety of the training process, and its multi-head attention layer learns proper transformations for the experts to align with the image features, mitigating the impact of the noise-contaminated prompts.

## C.3 ATTENTION-BASED VS. LINEAR PROJECTION-BASE GATING NETWORK

Table 7: Comparison between the proposed attention-based gating network and the linear projection-based gating network

| | Flowers102 | OxfordPets | Food101 | Caltech101 | DTD |
|---|---|---|---|---|---|
| Linear projection-based (3 experts) | 86.92±1.84 | 90.54±1.33 | 78.19±3.07 | 89.59±1.46 | 61.42±5.43 |
| Linear projection-based (10 experts) | 69.64±4.57 | 52.78±6.88 | 77.39±3.29 | 86.57±1.96 | 30.42±7.14 |
| Attention-based, with aggregation | 97.56±0.07 | 98.24±0.12 | 91.89±0.19 | 96.17±0.18 | 87.52±0.69 |
| Attention-based, without aggregation (ours) | 98.41±0.04 | 99.06±0.09 | 93.39±0.09 | 97.95±0.07 | 89.13±0.54 |

We compare the proposed attention-based gating network against the traditional linear projection-based gating network. The results are reported in Tab. 7 which strongly favor the attention-based approach, showing significantly better and more stable performance across all datasets. Notably, the linear projection-based approach struggles with larger numbers of experts (10 experts vs 3 experts), while the attention-based method maintains high performance. This is because the attention-based gating network 1) is more robust to adaptive experts; 2) serves as linear probing with more capacity; 3) leverages CLIP's feature alignment with attention mechanism; and 4) is agnostic to experts' order (detailed strength of attention-based approach over linear projection is described in Sec. 3.3).

In addition, we choose to avoid aggregating the gating network due to its larger size. The gating network can be magnitudes larger than the prompts (see Tab. 9), which would largely increase the communication overhead. This design choice also aims for higher performance. As the deepest parameterized module of the entire model, the gating network should be fully "personalized" as opposed to "globalized" to achieve higher performance (Arivazhagan et al., 2019; Collins et al., 2021). The process of aggregating the gating network limits the level of its personalization, which yields worse results than our method that keeps the gating network locally (see Tab. 7).

## C.4   ADDITIONAL ABLATION STUDY

Table 8: Ablation study: impact of number of experts ($K$ non-local experts + 1 local experts) to `pFedMoAP` on DomainNet

|  | Clipart | Infograph | Painting | Quickdraw | Real | Sketch | Average |
|---|---|---|---|---|---|---|---|
| $K+1=5$ | 47.20±0.30 | 46.80±0.48 | 32.54±0.42 | 37.67±0.52 | 31.50±0.65 | 36.09±0.92 | 38.63±0.29 |
| $K+1=10$ | 46.89±1.07 | 46.15±0.95 | 32.76±0.42 | 37.70±1.02 | 31.94±0.83 | 36.84±1.21 | 38.71±0.45 |
| $K+1=15$ | 47.36±1.12 | 46.29±0.83 | 32.76±0.57 | 37.76±0.98 | 31.78±0.72 | 36.67±1.03 | 38.77±0.38 |
| $K+1=20$ | 47.49±0.64 | 46.73±0.71 | 32.74±0.84 | 37.16±0.34 | 31.02±0.59 | 37.67±0.72 | 38.80±0.11 |
| $K+1=25$ | 47.56±0.92 | 46.89±1.30 | 32.23±0.97 | 38.03±1.14 | 31.66±0.77 | 36.79±1.02 | 38.86±0.43 |
| $K+1=30$ | 47.56±0.87 | 46.64±1.38 | 32.19±0.89 | 37.93±1.22 | 31.80±0.95 | 37.01±1.08 | 38.85±0.43 |

**Number of experts.** We present an ablation study examining how the number of experts ($K$ non-local experts + 1 local expert) affects `pFedMoAP`'s performance on the DomainNet dataset. The results in Tab. 8 show that the average performance gradually improves as the number of experts increases from 5 to 30. Fig. 5 plots the trend of performance with different numbers of experts from 5 to 40 on CIFAR10. For both datasets, it plateaus around larger numbers of experts, suggesting that while having multiple experts is beneficial, there's a point of diminishing returns. This is because the extra downloaded experts (selected last by our KNN-based expert selection mechanism) by a client are trained from the data that is distinctly distributed from the client's local data distribution. This shows that the proposed method does not require an excessive number of experts to achieve high performance.

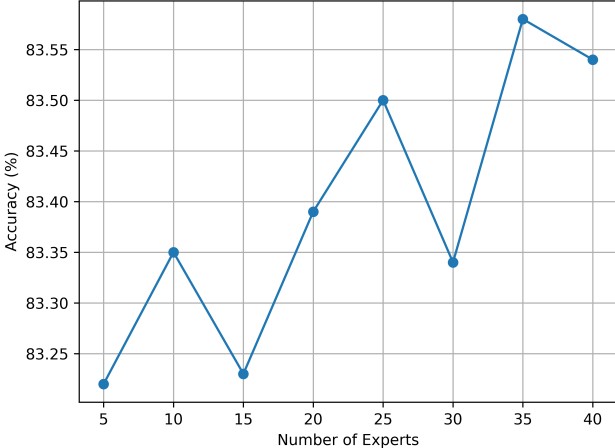

Figure 5: The impact of the number of experts on CIFAR10 with 100 clients

Table 9: Ablation study: the impact of output dimension of CLIP backbone to `pFedMoAP` on CLIP datasets. For $d_{\text{feature}} < 1024$, a pooling layer is added after the $d_{\text{feature}} = 1024$ feature from the backbone to reduce the size of the gating network as mentioned in Sec. 3.3

|  | Gating network size | Flowers102 | OxfordPets | Food101 | Caltech101 | DTD |
|---|---|---|---|---|---|---|
| $d_{\text{feature}} = 32$ | 4.2K | 97.28±0.18 | 98.75±0.32 | 93.42±0.08 | 97.37±0.08 | 88.61±0.89 |
| $d_{\text{feature}} = 64$ | 16.6K | 98.55±0.10 | 98.91±0.23 | 93.89±0.12 | 97.75±0.12 | 89.96±0.09 |
| $d_{\text{feature}} = 128$ | 66.0K | 98.41±0.04 | 99.06±0.09 | 93.39±0.09 | 97.95±0.07 | 89.13±0.54 |
| $d_{\text{feature}} = 256$ | 263.2K | 99.01±0.05 | 98.88±0.21 | 92.49±0.20 | 97.93±0.07 | 90.88±0.16 |
| $d_{\text{feature}} = 512$ | 1.1M | 98.18±0.38 | 96.85±0.22 | 90.34±0.31 | 96.99±0.11 | 89.65±0.10 |
| $d_{\text{feature}} = 1024$ | 4.2M | 98.11±0.33 | 95.81±0.84 | 89.20±0.37 | 96.82±0.26 | 89.03±0.14 |

**Feature dimension.** We investigate the impact of the output dimension ($d_{\text{feature}}$) of the CLIP backbone on `pFedMoAP`'s performance across CLIP datasets. The study shows that moderate feature dimensions (128-256) generally yield the best results, with $d_{\text{feature}} = 128$ or 256 consistently performing well across all datasets. Lower feature dimensions ($d_{\text{feature}} = 32$ or 64) have less capacity, failing to unlock the full potential of the pre-trained VLM. High dimensions ($d_{\text{feature}} = 128$ or 1024) significantly increase the size of the gating network. With less training data on a client, high dimensions lead to overparameterized gating network, resulting in slightly degraded performance. This suggests that a relatively compact feature dimension is sufficient and possibly preferable for the attention-based gating mechanism.

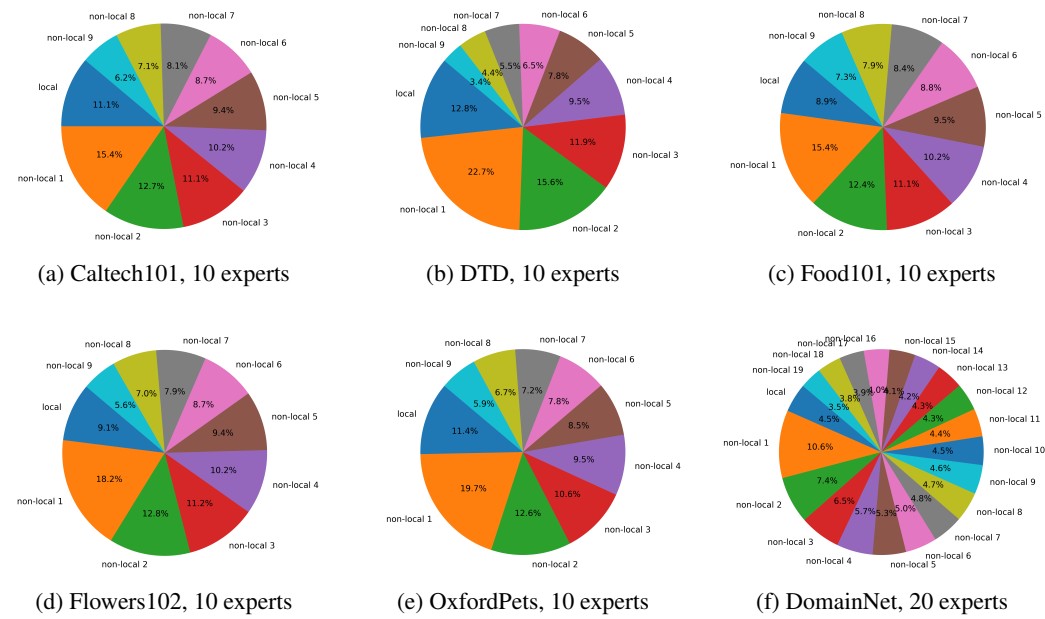

(a) Caltech101, 10 experts  (b) DTD, 10 experts  (c) Food101, 10 experts

(d) Flowers102, 10 experts  (e) OxfordPets, 10 experts  (f) DomainNet, 20 experts

Figure 6: Contribution of the experts based on averaged attention score across all test images. The first five charts are for CLIP datasets, for which there are 10 clients in each dataset. The last chart is for DomainNet with a total of 30 clients.

**Visualization of contribution for each expert.** To have a clear understanding of how the experts contribute to the MoE text feature ($\boldsymbol{T}_{MoE}$), we visualize the contribution of the local and non-local experts towards it. This is achieved by taking the following steps. For each image in a client's test set, we compute the attention score (softmax of scaled dot product between the image and text features (Vaswani, 2017)) for each expert (local and non-local) through the multi-head attention layer in the proposed gating network. The scores from non-local experts are sorted in a descending manner. All scores are averaged over the test set across all clients. We visualize the averaged scores under 6 datasets in Fig. 6. From this figure, we can see that **1)** every expert has a non-trivial contribution to $\boldsymbol{T}_{MoE}$, the text feature generated from the MoE, indicating that each expert is indispensable in the whole design; 2) as a result of the minimization, the contribution from the local experts to the MoE is not the highest. This is reasonable since besides the $\boldsymbol{T}_{MoE}$, Eq. (10)'s second term optimizes the

text feature from the local expert itself ($T_L$) based on the image feature, which directly contributes to the loss; and 3) although not the highest, the contribution from the local expert to the MoE is still among the highest ones as it is the only expert that is updated during local training with fixed non-local prompts.

