# OpenReview forum: "Mixture of Experts Made Personalized: Federated Prompt Learning for Vision-Language Models"
_ICLR.cc/2025/Conference — ICLR 2025 Poster_

### Official Review · Reviewer_iqPq · 2024-10-31

**Soundness:** 3
**Presentation:** 2
**Contribution:** 2
**Rating:** 6
**Confidence:** 3

**Summary:**

This work introduces a new paradigm in federated prompt learning for pre-trained Vision-Language Models (VLMs) such as CLIP, challenging the traditional restriction that clients are limited to downloading a single globally aggregated model. This shift is particularly beneficial for lightweight prompts, enabling more flexible and effective knowledge sharing among clients.

**Strengths:**

1. pFedMoAP proposes a novel framework that personalizes the prompt learning process by allowing clients to download multiple pre-aggregated prompts as fixed non-local experts. This leverages the concept of Mixture of Experts, where a local attention-based gating network is implemented to generate enhanced text features that better align with the local image data on the client side.

2. By facilitating a many-expert scenario (e.g., 10 experts) per client with negligible communication overhead, pFedMoAP achieves efficient and effective personalization. This approach overcomes the limitations of having too many or too few experts, which can lead to high communication costs or suboptimal performance, respectively.

3. The algorithm consistently outperforms existing work across various heterogeneous federated settings, as demonstrated through extensive experiments on 9 datasets.

**Weaknesses:**

1. Determining which prompt experts should be included in the pool and when to update or replace them requires careful management. If the selection process is not optimal, it could lead to suboptimal performance. Additionally, maintaining a diverse and up-to-date pool of experts is crucial but can be complex.

2. The local attention-based gating network, while designed to be efficient, may still introduce optimization challenges. Ensuring that this network converges to a solution that effectively combines local and non-local prompts is not trivial, especially when dealing with heterogeneous data distributions.

2.  As the number of clients increases, managing a dynamic pool of prompt experts and maintaining efficient communication between the server and all clients can become more challenging. The complexity of coordinating updates and ensuring that each client receives the most relevant prompts could grow significantly.

**Questions:**

Please see weaknesses.

---

> ### Author Response · Authors · 2024-11-24
> **Reply to Reviewer iqPq**
>
> We appreciate the thoughtful feedback provided by the reviewers. Below we address each concern in detail:
>
> **Reply to W1.** Our approach incorporates several design choices that make this process both efficient and effective:
>
> - Our KNN-based expert selection method (Section 3.2) provides an automatic, data-driven approach to expert inclusion that adapts to the distribution of client data. This removes the need for manual intervention in expert selection.
>
> - The pool maintains natural diversity through our dynamic refresh mechanism (Eq. (7)), which ensures the pool stays current with the latest client updates while preserving historical knowledge through non-participating clients' prompts.
>
> - The lightweight nature of prompts (only 8,192 parameters per prompt as noted in Sec. 1) makes maintaining even a large pool computationally feasible, allowing us to keep more prompts in the pool rather than aggressively removing them.
>
> While finding the theoretically optimal selection method is impractical and beyond the scope of our work, our extensive experiments across 9 datasets demonstrate that the proposed approach consistently outperforms state-of-the-art alternatives, validating the effectiveness of our pool management strategy.
>
> **Reply to W2.** Our design tries to specifically address the optimization challenge under data heterogeneity concern by the following strategies:
>
> - The parameter-efficient design (Section 3.3) reduces the network from 4.2M to just 66.0K parameters through strategic dimension reduction, making optimization more tractable.
>
> - Our multi-head attention mechanism learns relationships between image and text features rather than direct expert weights ($G(\cdot)_i$), providing greater robustness to heterogeneous data (as explained in Section 3.3) by leveraging the feature alignment ability of pre-trained CLIP-like VLMs.
>
> - The weighted averaging approach (Equation 10) ensures stable training by maintaining a balance between local expertise and collective knowledge.
>
> These design choices result in stable convergence, as evidenced by our experimental results across diverse federated settings.
>
> **Reply to W3.** We agree with the reviewer that the proposed method increases the complexity of server maintenance complexity, and thus might not be a perfect fit for extremely large number of clients (e.g. 1 million clients). This being said, due to the compact size of prompts (8,192 parameters), both communication costs and server-side pool management remain efficient. Our approach maintains this efficiency even with large numbers of clients for several reasons:
>
> - The extremely small size of prompts (8,192 parameters) means communication overhead remains minimal even with many clients. This is orders of magnitude more efficient than traditional federated learning approaches that share full models.
>
> - Our KNN-based expert selection automatically sets a limit to the maximum transmitted prompts and the communication overhead, ensuring communication complexity scales linearly rather than quadratically with the number of clients.
>
> - The server-side pool management is computationally efficient, requiring only simple matrix operations for updates and KNN calculations.
>
> Furthermore, we would like to mention that the benefits of improved personalization and performance outweigh the modest additional coordination overhead, as demonstrated by our comprehensive experimental results.
>
> Please feel free to let us know if you have any more concerns, and we will try our best to address them. Thank you for your feedback.

---

### Official Review · Reviewer_CcjC · 2024-11-02

**Soundness:** 3
**Presentation:** 3
**Contribution:** 3
**Rating:** 6
**Confidence:** 5

**Summary:**

This paper proposes a Mixture of Experts (MoE) approach for prompt-based federated learning, employing a local attention-based gating network to refine text features for better alignment with local image data. It also highlights the lightweight design of the prompt, exploring a configuration where the prompt is updated directly by other clients rather than relying on an aggregated prompt from the server.

**Strengths:**

1. The integration of the MoE method into prompt-based federated learning is an innovative concept.

2. The paper is well-organized and effectively communicates its main contributions. The methodology is clearly explained, particularly regarding the incorporation of the MoE approach in the federated learning context.

**Weaknesses:**

1. While this approach utilizes inter-client sharing, it could expose prompt updates directly to other clients, which may lead to privacy concerns as prompt updates could be tracked, making the model susceptible to certain attack algorithms.

2. In the related work section in the appendix (Line 783), the authors mentioned that previous work utilized inter-client sharing prior to aggregation. However, these works allow prompt aggregation in the server and do not share local prompts with other clients. This sentence is somewhat unclear; please provide a clearer version to avoid misunderstandings.

3. The paper claims that the MoE-based method can leverage prompts from other clients. Please provide experimental evidence to show whether your method’s effectiveness is sensitive to the total number of clients and the value of $K$.

4. Some related works are encouraged to be properly cited in Related work. For example, personalized federated learning [a,b] and prompt-based federated Learning [c].

[a] Li, et al, FedTP: Federated Learning by Transformer Personalization, TNNLS 2023.

[b] Cai, et al, Fed-CO2: Cooperation of Online and Offline Models for Severe Data Heterogeneity in Federated Learning, NeurIPS 2023.

[c] Pan et al, Federated Learning from Vision-Language Foundation Models: Theoretical Analysis and Method, NeurIPS 2024.

**Questions:**

Please refer to the weakness.

---

> ### Author Response · Authors · 2024-11-24
> **Reply to Reviewer CcjC**
>
> We would like to thank the reviewer for your time and patience in reviewing our manuscript. Below, we reply to every concern in weaknesses and questions.
>
>
> **Reply to W1.** We conduct additional user-level differential privacy experiments (line 858-887 & Tab. 5). Below is a preview.
>
> **Table 5: Performance under ($\epsilon$, $\delta$)-differential privacy on CLIP datasets under pathological non-IID setting.**
>
> | Method | Flowers102 | OxfordPets | Food101 | Caltech101 | DTD |
> |--------|------------|------------|----------|------------|-----|
> | **Without differential privacy (from Tab. 1)** |||||
> | PromptFL (Guo et al., 2023b) | 72.80±1.14 | 90.79±0.61 | 77.31±1.64 | 89.70±1.99 | 54.11±0.22 |
> | PromptFL+FedProx (Li et al., 2020b) | 66.40±0.29 | 89.24±0.41 | 76.24±1.94 | 89.41±0.55 | 44.26±1.11 |
> | pFedMoAP (ours) |98.41±0.04 | 99.06±0.09 | 93.39±0.09 | 97.95±0.07 | 89.13±0.54 |
> | **With differential privacy (ε = 50)** |||||
> | PromptFL (Guo et al., 2023b) |67.07±0.60 | 88.05±0.32 | 77.41±0.60 | 84.83±0.42 | 38.39±1.25 |
> | PromptFL+FedProx (Li et al., 2020b) | 66.22±0.63 | 87.78±0.61 | 77.27±0.59 | 84.68±0.64 | 39.43±1.11 |
> | pFedMoAP (ours) |98.34±0.06 | 99.08±0.02 | 93.36±0.04 | 97.90±0.08 | 89.99±0.49 |
> | **With differential privacy (ε = 25)** |||||
> | PromptFL (Guo et al., 2023b) |64.25±1.10 | 86.26±1.07 | 76.84±0.66 | 85.00±1.59 | 38.19±0.66 |
> | PromptFL+FedProx (Li et al., 2020b) |62.87±0.99 | 86.82±0.47 | 76.21±0.64 | 84.51±1.52 | 37.82±0.52 |
> | pFedMoAP (ours) | 98.36±0.12 | 99.02±0.04 | 93.41±0.13 | 97.99±0.06 | 89.11±0.28 |
>
>
>
>
>
>
> And we would also like to mention that in pFedMoAP, clients remain unaware of the origin of non-local prompts, including which client they come from or the training round they were last updated in. Besides, the non-local prompts are downloaded after a selection process on the server. These factors collectively make it extremely difficult for a client to infer the gradient associated with a prompt or the data it has been trained with, which largely mitigates the privacy risk.
>
>
>
> **Reply to W2.** We have modified the description of the previous works in line 783 to avoid misunderstanding. Essentially, what we wanted to express is that the previous existing prompt-based FL works habitually adhere to the paradigm where locally learned prompts are not allowed for inter-client sharing before aggregation. This is because these works are inherent in the traditional FL framework for full-size models (as opposed to the prompts). Sharing full-size model before aggregation will cause unaffordable communication overhead, and thus, traditional FL does not consider inter-client sharing. However, since prompt is much smaller than the full-size model, this inheritance of the paradigm becomes unnecessary.

---

> > ### Author Response · Authors · 2024-11-24
> > **Reply to Reviewer CcjC (cont'd)**
> >
> > **Reply to W3.** We have added the ablation study on the number of experts (line 911-930 & Tab. 7). Below is a preview.
> >
> > **Table 7: Ablation study: impact of number of experts ($K$ non-local experts + $1$ local experts) to pFedMoAP on DomainNet**
> >
> > |       | Clipart | Infograph | Painting | Quickdraw | Real | Sketch | Average |
> > |-------|----------|-----------|-----------|-----------|------|---------|---------|
> > | $K + 1=5$  |47.20±0.30 | 46.80±0.48 | 32.54±0.42 | 37.67±0.52 | 31.50±0.65 | 36.09±0.92 | 38.63±0.29 |
> > | $K + 1=10$ |46.89±1.07 | 46.15±0.95 | 32.76±0.42 | 37.70±1.02 | 31.94±0.83 | 36.84±1.21 | 38.71±0.45 |
> > | $K + 1=15$ |47.36±1.12 | 46.29±0.83 | 32.76±0.57 | 37.76±0.98 | 31.78±0.72 | 36.67±1.03 | 38.77±0.38 |
> > | $K + 1=20$ |47.49±0.64 | 46.73±0.71 | 32.74±0.84 | 37.16±0.34 | 31.02±0.59 | 37.67±0.72 | 38.80±0.11 |
> > | $K + 1=25$ |47.56±0.92 | 46.89±1.30 | 32.23±0.97 | 38.03±1.14 | 31.66±0.77 | 36.79±1.02 | 38.86±0.43 |
> > | $K + 1=30$ |47.56±0.87 | 46.64±1.38 | 32.19±0.89 | 37.93±1.22 | 31.80±0.95 | 37.01±1.08 | 38.85±0.43 |
> >
> >
> > We would also like to sincerely apologize for a typo in line 415 (previous version: $N=10$, corrected version: $N=100$) for CIFAR10 and CIFAR100. This typo has been corrected. The table below summarizes our experimental settings regarding the number of clients.
> >
> > | Dataset           | Training Set Size | Test Set Size | Number of Classes | Number of Clients | Sample Rate | Data Heterogeneity      |
> > |-------------------|-------------------|---------------|-------------------|-------------------|-------------|-------------------------|
> > | Flowers102        | 4,093             | 2,463         | 102               | 10                | 100%        | Pathological non-IID    |
> > | OxfordPets        | 2,944             | 3,669         | 37                | 10                | 100%        | Pathological non-IID    |
> > | Food101           | 50,500            | 30,300        | 101               | 10                | 100%        | Pathological non-IID    |
> > | Caltech101        | 4,128             | 2,465         | 100               | 10                | 100%        | Pathological non-IID    |
> > | DTD               | 2,820             | 1,692         | 47                | 10                | 100%        | Pathological non-IID    |
> > | Office-Caltech10  | 2,025             | 508           | 10                | 20                | 50%         | Dir(0.3)                |
> > | DomainNet         | 18,278            | 4,573         | 10                | 30                | 25%         | Dir(0.3)                |
> > | CIFAR10           | 50,000            | 10,000        | 10                | 100               | 10%         | Dir(0.5)                |
> > | CIFAR100          | 50,000            | 10,000        | 100               | 100               | 10%         | Dir(0.5)                |
> >
> >
> >
> > **Reply to W4.** Thank you for the suggestion. We have included the mentioned related works in the latest version of the manuscript.
> >
> >
> > Please feel free to let us know if you have any more concerns, and we will try our best to address them. Thank you for your feedback.

---

> > ### Comment · Reviewer_CcjC · 2024-11-25
> > **Comment by Reviewer CcjC**
> >
> > Thanks for the responses and revisions. Most of the concerns have been addressed.
> >
> > I am happy to see the additional experiments with differential privacy. My previous concerns about prompt sharing's privacy issues have been alleviated. If privacy is not an issue, then enabling prompt sharing among clients within the Federated Prompt Learning framework is a good idea.
> >
> > Regarding the additional experiments with different numbers of experts $K$, it seems that the final performance is not quite related to $K$. Please analyze this phenomenon.

---

> > > ### Author Response · Authors · 2024-11-28
> > >
> > > Thank you for your feedback. For the impact of the number of experts on the performance, since the non-local experts are selected for a client based on a KNN sparse selection algorithm (Sec. 3.2), the top beneficial non-local experts are already selected even with smaller number of experts (e.g. $K+1=5$ in Tab. 7), whereas larger number of experts (e.g. $K+1=25$ or $30$ in Tab. 7) can introduce redundancy or extra experts that are not highly beneficial with corresponding data distributions further away from the local client. Additionally, since both severe feature shift and label shift are presented in DomainNet, such a situation might be exacerbated (line 416-419, Implementation details). We also conducted **additional experiments (line 946)** on CIFAR10 (Dir($\alpha$=0.5) with 100 clients), where only label shift is presented (Fig. 5, line 946). We can see from the results that a clear positive correlation between the performance and the number of experts is shown in the plot.
> > >
> > > Please feel free to let us know if you have any more concerns. And we appreciate your consideration for raising the score. Thank you.

---

> > > > ### Comment · Reviewer_CcjC · 2024-12-03
> > > > **Comment by Reviewer CcjC**
> > > >
> > > > Thanks for the response. I have no further concerns. Together with other reviewers' comments, I will maintain my positive score.

---

> > > ### Author Response · Authors · 2024-11-30
> > >
> > > Just a gentle reminder that there are only a few days left for the discussion. We would like to ask if the reviewer has any more concerns/questions. We will strive to address every concern and clarify each question in details.
> > >
> > > We sincerely appreciate the reviewer's consideration on raising the score if there is no more concern/question. Thank you.

---

> > > ### Author Response · Authors · 2024-12-02
> > >
> > > Dear reviewer CcjC,
> > >
> > > Thanks again for your feedback. As the discussion period is concluding within 24 hours for the reviewers, could you please check our response and let us know if anything remains unclear?
> > >
> > > If your concerns are resolved, we appreciate your consideration of increasing the score.
> > >
> > > Thank you very much.
> > >
> > > Best,
> > > Authors

---

### Official Review · Reviewer_zSUv · 2024-11-03

**Soundness:** 2
**Presentation:** 1
**Contribution:** 2
**Rating:** 5
**Confidence:** 4

**Summary:**

This paper proposes a lightweight federated prompt learning framework to personalize the prompt learning process through the lens of mixture of experts (MoE). An attention-based gating network is also introduced to achieve efficient cross-client knowledge sharing. Experiments indicate that the proposed **pFedMoAP** performs better than state-of-the-art methods.

**Strengths:**

1. Implementing prompt learning into the distributed environment is an important topic for FL applications due to its efficiency in computation and communication.

2. The proposed learning framework is simple and effective.

**Weaknesses:**

1. The quality of the writing is poor. The expression of the paper is somehow obscure, and not concise enough. Besides, there are many typos in the paper, e.g.,  lacking space between two adjacent words in Line 27, Line 198, Lin 415, and Line 469.

2. The novelty is limited. The **pFedMoAP** seems like a naive combination of PromptFL and MoE. Although the selected non-local experts will be updated in each communication round, the intuition behind it is also not explained well, leading to weak convincingness.

3. it is unclear why the attention-based gate network does not need to be uploaded to the server for aggregation. Please clarify.

4. I think the ablation study of the paper is not enough.
- To study the effectiveness of MoE-based textual feature enhancement mechanism, the authors should add experiments under different values of $K$ to confirm performance changes.

- The author implements a dimension reduction operation in Section 3.3, please add experiments to show whether the performance will be influenced.

- How about a larger number of clients, e.g., 100 or more?

**Questions:**

1. In Section 4.1 **Implementation details.**, for CIFAR10&CIFAR100, why the participation rate is 10%? Does it mean only one client will be selected to join the training process in each communication round?

---

> ### Author Response · Authors · 2024-11-24
> **Reply to Reviewer zSUv**
>
> We would like to thank the reviewer for your time and patience in reviewing our manuscript. Below, we reply to every concern in weaknesses and questions.
>
> **Reply to W1.** We sincerely apologize for the typos and lack of spaces between words. We have corrected them in the latest version.
>
> **Reply to W2a.** We agree that pFedMoAP is under the umbrella of prompt-based FL with MoE (the synergy of which has not been explored by the community). Meanwhile, we also believe that this fact does not cover up our novelty and contributions, because:
> - to the best of our knowledge, we are the first to propose a Mixture of Adaptive Prompt architecture for CLIP-like VLMs under federated settings. The proposed pipeline (Fig. 2) efficiently brings non-local knowledge in the adaptive prompts to boost the client's performance.
> - we circumvent the paradigm of traditional FL with the pre-aggregated weights are not allowed to share. Existing prompt-based FL works habitually adhere to this paradigm, failing to fully leverage the lightweight nature of prompt.
> - most importantly, from pure MoE perspective, our proposed attention-based gating network for VLMs is novel and favorable compared to the the linear projection-based gating network in traditional MoE. We detailed the reasons of it under federated settings in line: line 316-353. Briefly, attention-based gating network 1) is more robust to adaptive experts; 2) serves as linear probing with more capacity; 3) leverages CLIP's feature alignment with attention mechanism; and 4) is agnostic to experts' order. We conduct additional experiments to compare these two types of gating network to show the superiority of the proposed attention-based gating network. *(line 889-909 & Tab. 6)*
>
> **Reply to W2b.** The intuition behind updating the non-local experts is that as the training progresses, the trained prompts become more and more like *experts* as they converge. The training can be considered as a process that keeps embedding various client-specific information in the prompts. When downloaded to a client, the more developed and thoroughly trained prompts offer more non-local information and help the local training as more qualified experts.
>
> **Reply to W3.** The main reason why the gating network is not aggregated is due to the communication overhead. We provide the sizes of different gating network (Tab. 8) based on different (pooled) dimensions of the feature output by CLIP's image and text encoders. We can see that the size of the gating network in general is significantly larger than a prompt with size $16\times512\approx 8k$. Uploading and downloading the gating network will increase the communication overhead by a magnitude or more. Besides, we conduct additional experiments *(line 889-909 & Tab. 6)* and empirically show that aggregating the gating network would not much benefit the personalized performance. This shows that the global aggregating with a short personalization (i.e. the selected round) of the gating network can hardly work better than the completely localized gating network.

---

> > ### Author Response · Authors · 2024-11-24
> > **Reply to Reviewer zSUv (cont'd)**
> >
> > **Reply to W4a, W4b.**  Thank you for your advice. We conduct additional ablation study on the number of experts and the dimension of features. Below is a preview of the results. More details can be found in line 911-950.
> >
> > **Table 7: Ablation study: impact of number of experts ($K$ non-local experts + $1$ local experts) to pFedMoAP on DomainNet**
> >
> > |       | Clipart | Infograph | Painting | Quickdraw | Real | Sketch | Average |
> > |-------|----------|-----------|-----------|-----------|------|---------|---------|
> > | $K + 1=5$  |47.20±0.30 | 46.80±0.48 | 32.54±0.42 | 37.67±0.52 | 31.50±0.65 | 36.09±0.92 | 38.63±0.29 |
> > | $K + 1=10$ |46.89±1.07 | 46.15±0.95 | 32.76±0.42 | 37.70±1.02 | 31.94±0.83 | 36.84±1.21 | 38.71±0.45 |
> > | $K + 1=15$ |47.36±1.12 | 46.29±0.83 | 32.76±0.57 | 37.76±0.98 | 31.78±0.72 | 36.67±1.03 | 38.77±0.38 |
> > | $K + 1=20$ |47.49±0.64 | 46.73±0.71 | 32.74±0.84 | 37.16±0.34 | 31.02±0.59 | 37.67±0.72 | 38.80±0.11 |
> > | $K + 1=25$ |47.56±0.92 | 46.89±1.30 | 32.23±0.97 | 38.03±1.14 | 31.66±0.77 | 36.79±1.02 | 38.86±0.43 |
> > | $K + 1=30$ |47.56±0.87 | 46.64±1.38 | 32.19±0.89 | 37.93±1.22 | 31.80±0.95 | 37.01±1.08 | 38.85±0.43 |
> >
> >
> > **Table 8: Ablation study: the impact of output dimension of CLIP backbone to pFedMoAP on CLIP datasets. For $d_{\text{feature}}$ < 1024, a pooling layer is added after the $d_{\text{feature}} = 1024$ feature from the backbone to reduce the size of the gating network as mentioned in Sec. 3.3**
> >
> > |  | Gating network size | Flowers102 | OxfordPets | Food101 | Caltech101 | DTD |
> > |-----------|-------------------|------------|------------|----------|------------|-----|
> > | $d_{\text{feature}} = 32$ | 4.2K | 97.28±0.18 | 98.75±0.32 | 93.42±0.08 | 97.37±0.08 | 88.61±0.89 |
> > | $d_{\text{feature}} = 64$ | 16.6K | 98.55±0.10 | 98.91±0.23 | 93.89±0.12 | 97.75±0.12 | 89.96±0.09 |
> > | $d_{\text{feature}} = 128$ | 66.0K |98.41±0.04 | 99.06±0.09 | 93.39±0.09 | 97.95±0.07 | 89.13±0.54 |
> > | $d_{\text{feature}} = 256$ | 263.2K |99.01±0.05 | 98.88±0.21 | 92.49±0.20 | 97.93±0.07 | 90.88±0.16 |
> > | $d_{\text{feature}} = 512$ | 1.1M | 98.18±0.38 | 96.85±0.22 | 90.34±0.31 | 96.99±0.11 | 89.65±0.10 |
> > | $d_{\text{feature}} = 1024$ | 4.2M | 98.11±0.33 | 95.81±0.84 | 89.20±0.37 | 96.82±0.26 | 89.03±0.14 |
> >
> >
> > **Reply to W4c, Q1.** We sincerely apologize for the typo. The number of clients in all CIFAR10 and CIFAR100 experiments are *100, instead of 10* (line 415, Implementation details of Sec. 4.1). The sample rate is 10% (so 10 clients are sampled for each round). The purpose of this setting is to use more clients and less sample rate to simulate a cross-device setting. We apologize for the misunderstanding due to this typo.
> >
> > Please feel free to let us know if you have any more concerns, and we will try our best to address them. Thank you for your feedback.

---

> > > ### Comment · Reviewer_zSUv · 2024-11-25
> > >
> > > Thank the authors for the explanation and additional experiments. Results in C.3 look good, but I am still concerned about the design of the optional expert group (i.e., aggregated prompt and other local prompts), which makes the novelty not convincing enough. I am curious about, the local prompt and other non-local pre-aggregated prompts, who has a greater influence on the personalized performance? It is suggested to visualize the attention score of each expert to indicate that each expert/prompt is indispensable in the whole design.

---

> > > ### Comment · Reviewer_zSUv · 2024-11-25
> > >
> > > Besides, I am confused with the results in C.2, why is the performance of "**Attention-based, with aggregation**" worse than "**Attention-based, without aggregation (ours)**"？This seems counter-intuitive, a more detailed explanation would be beneficial to verify the design of pFedMoAP.

---

> ### Author Response · Authors · 2024-11-28
>
> Thank you for your feedback and suggestions on the visualization, for which we added a **new set of experiments (line 969-1006 and Fig.6)**. In these experiments, we visualize the contribution of the local and non-local experts towards the MoE on 6 datasets. In addition, we also conduct **new experiments (Fig. 5, line 946)** and plot the trend of the performance with different numbers of experts on CIFAR10 (Dir($\alpha$=0.5) with 100 clients). Both experiments suggest that the non-local experts are indispensable to the performance gain. Meanwhile, as the incorporation of the non-local experts has been empirically proven in these experiments, we would also like to politely mention that the other reviewers have all given credit to our novelty, and we briefly summarize our three major contributions/novelties as 1) the algorithm that personalizes prompt-based federated learning for VLMs with MoE architecture; 2) the attention-based gating network for VLM (not necessarily for FL) which outperforms the traditional projection-based gating network by large margins; and 3) the paradigm shift from existing FL approaches that limit pre-aggregated models from inter-client sharing.
>
> The reason why the performance of aggregated gating is worse than non-aggregated gating is that, while aggregating the gating network embeds global knowledge to the parameter weights, as the deepest layer of the entire model that is parameterized, it should be fully "personalized" as opposed to "globalized" to achieve higher performance [1, 2]. The process of aggregating the gating network limits the extent of personalization to only one round of personalization (the round where the client is selected). This is much less than fixing the gating network locally throughout the entire federated training, especially in the case of few-shot learning (Tab. 6's results are on CLIP datasets, which are used for few-shot learning as mentioned in line 373) where one round of fine-tuning involves few images for the gating network to be trained with, hence the worse performance. With this being said, the design choice of not aggregating the attention-based gating network is also due to the large size of the gating network (Tab. 8). The attention-based network can be up to around 3 magnitudes larger than the prompts (gating with $d_{\text{feature}}=1024$: 4.2M parameters, prompt:16$\times$512 parameters), which would largely increase the communication overhead.
>
>
> Please feel free to let us know if you have any more concerns. And we appreciate your consideration for raising the score. Thank you.
>
> [1] Federated Learning with Personalization Layers [AISTATS 2020]
>
> [2] Exploiting Shared Representations for Personalized Federated Learning [ICML 2021]

---

> > ### Comment · Reviewer_zSUv · 2024-11-28
> >
> > We thank the authors for their hard work in addressing my concerns.  I would suggest incorporating the discussions into the revision of this paper, especially for the explanation of why the performance of aggregated gating is worse than non-aggregated gating. Upon reading the rebuttal and other's reviews, I have increased my rating.

---

> > > ### Author Response · Authors · 2024-11-28
> > >
> > > We appreciate the reviewer's suggestions and thank the reviewer for increasing the rating. We will incorporate the discussions into the revision of this paper.
> > >
> > > We would also like to kindly ask if the reviewer has any more concerns/questions. We will strive to address every concern and clarify each question in details. We wish the reviewer could change to a positive rating in case there is no more concern/question.
> > >
> > > Happy Thanksgiving :)

---

### Official Review · Reviewer_eFjf · 2024-11-03

**Soundness:** 3
**Presentation:** 3
**Contribution:** 3
**Rating:** 6
**Confidence:** 2

**Summary:**

This paper introduces pFedMoAP that enables effective federated prompt learning for vision-language models like CLIP. The key innovation is allowing clients to download multiple pre-aggregated prompts as fixed non-local experts rather than being restricted to a single globally aggregated model.

**Strengths:**

1.	The method is novel which allows the client download prompts from others.
2.	The experiments are extensive and show good performance.

**Weaknesses:**

1.	It is very similar to the prompt learning framework. How does your method differ from the classic prompt learning framework if you can directly access prompts from other clients?
2.	I noticed that similar work, like pFedPrompt, conducted experiments on large-scale datasets like UCF-101 or ImageNet. Would the scale of the dataset influence the performance?

**Questions:**

Refer to the weakness part.

---

> ### Author Response · Authors · 2024-11-24
> **Reply to Reviewer eFjf**
>
> We would like to thank the reviewer for your time and patience in reviewing our manuscript. Below, we reply to every concern in weaknesses and questions.
>
> **Reply to W1.** In pFedMoAP, the proposed mixture of adaptive prompt is different from classic prompt learning. Specifically, for Vision-Language Models (VLMs), classic prompt learning (e.g. CoOp [1]) does not involve mixture of expert prompt. On the other hand, in pFedMoAP
> - we proposed to treat the adaptive prompts as the experts
> - the prompts, trained from different distributions, are integrated by the proposed attention-based gating network, largely increasing the chance of better alignment for the output text feature with the image feature from CLIP-like VLMs (e.g. Table 1, pFedMoAP vs. CoOp). While CoOp only trains a local text feature ($T_L$), in pFedMoAP, we have $T_L$, $T_{NL_1}$, $T_{NL_2}$, ..., $T_{NL_K}$ with the proposed attention-based gating network to optimize for the alignment.
> - compared to traditional linear projection-based gating network, the novel attention-based gating network for CLIP-like VLMs is favorable due to 4 reasons detailed in Sec. 3.3, line 316-348. Briefly, attention-based gating network 1) is more robust to adaptive experts; 2) serves as linear probing with more capacity; 3) leverages CLIP's feature alignment with attention mechanism; and 4) is agnostic to experts' order. We conduct additional experiments to compare these two types of gating network to show the superiority of the proposed attention-based gating network (line 889-909).
>
> Besides, we would also like to mention that ''access prompts from other clients'' itself is a breakthrough from traditional prompt-based federated learning (FL) where existing algorithms only allow the aggregated prompt, instead of the raw pre-aggregated prompt, to be transmitted. This traditional paradigm unnecessarily embeds the global knowledge into the compressed aggregated prompt with loss of various local information, while pFedMoAP fully leverages the lightweight nature of the prompt and transmits the global knowledge in its raw form as the non-local adaptive prompt experts.
>
> **Reply to W2.** Our experiments follow most recent prompt-based FL for VLMs works' standard [2,3] and used all 9 datasets they have used. The table below lists the datasets we used and the partition method for each dataset. We regretfully do not have enough resources (time/GPU) to finish extra experiments on large-scale datasets within the discussion period and leave this as future work.
>
>
> | Dataset           | Training Set Size | Test Set Size | Number of Classes | Number of Clients | Sample Rate | Data Heterogeneity      |
> |-------------------|-------------------|---------------|-------------------|-------------------|-------------|-------------------------|
> | Flowers102        | 4,093             | 2,463         | 102               | 10                | 100%        | Pathological non-IID    |
> | OxfordPets        | 2,944             | 3,669         | 37                | 10                | 100%        | Pathological non-IID    |
> | Food101           | 50,500            | 30,300        | 101               | 10                | 100%        | Pathological non-IID    |
> | Caltech101        | 4,128             | 2,465         | 100               | 10                | 100%        | Pathological non-IID    |
> | DTD               | 2,820             | 1,692         | 47                | 10                | 100%        | Pathological non-IID    |
> | Office-Caltech10  | 2,025             | 508           | 10                | 20                | 50%         | Dir(0.3)                |
> | DomainNet         | 18,278            | 4,573         | 10                | 30                | 25%         | Dir(0.3)                |
> | CIFAR10           | 50,000            | 10,000        | 10                | 100               | 10%         | Dir(0.5)                |
> | CIFAR100          | 50,000            | 10,000        | 100               | 100               | 10%         | Dir(0.5)                |
>
> [1] Learning to Prompt for Vision-Language Models [IJCV 2022]
>
> [2] Global and Local Prompts Cooperation via Optimal Transport for Federated Learning [CVPR 2024]
>
> [3] Harmonizing Generalization and Personalization in Federated Prompt Learning [ICML 2024]
>
>
> Please feel free to let us know if you have any more concerns, and we will try our best to address them. Thank you for your feedback.

---

### Author Response · Authors · 2024-11-24
**Reply to all reviewers**

Dear reviewers,

We thank every reviewer for your time and patience to review our manuscript. We appreciate that the reviewers give credit to the novelty of our work (Reviewer eFjf, CcjC, iqPq), extensive experiments (Reviewer eFjf), importance of the topic (Reviewer zSUv), method effectiveness (Reviewer eFjf, zSUv, iqPq), clear paper organization and method explanation (Reviewer CcjC).

We have replied to your comments and addressed every concern that you have. Please do let us know if the concerns have been clearly addressed or if you have any other concerns.

Below, we list the major changes besides clarified descriptions and fixed typos in the updated pdf.
- The impact of differential privacy to the proposed method *(line 858-887 & Tab. 5)* -- pFedMoAP shows negligible performance degradation under strong privacy guarantee
- Comparison between the proposed attention-based gating network and the linear projection-based gating network *(line 889-909 & Tab. 6)* -- attention-based approach shows significantly better performances
- Ablation study: the impact of the number of experts to pFedMoAP *(line 911-930 & Tab. 7)* -- gradually performs better with more experts, plateaus at higher number of experts
- Ablation study: the impact of output dimension of CLIP backbone to pFedMoAP *(line 931-950 & Tab. 8)* -- larger dimension leads to overparameterized gating network, smaller leads to insufficient capacity.



We appreciate your feedback and consideration of increasing the rating. Thank you very much.

---

> ### Author Response · Authors · 2024-11-24
> **Easy tracking of modifications**
>
> For easy tracking of our major modifications (apart from small modifications such as typo fixing) in the updated pdf, we mark the sentences with major modifications in **blue** (one appearance, line 788-793). We also mark **blue** and **highlight** the titles of the added sections (Sec. C (line 854) and its subsections). Thank you for your time.

---

### Meta-Review · Area_Chair_ALRH · 2024-12-19

**Metareview:**

This paper introduces a lightweight federated prompt learning framework that uses a mixture of experts to personalize prompts and an attention-based gating network for efficient knowledge sharing across clients. It outperforms existing methods and allows direct updates of the prompts by clients.

There is some appreciation on the novel method allowing clients to download prompts from others, enhancing the personalization of the prompt learning in federated learning. The reviewers also recognize the proposed framework is simple, effective and achieve efficient computation and communication. Extensive experiments across 9 datasets show that the algorithm consistently outperforms existing methods in various setting.

While some reviewers share the concerns regarding the clarity and writing quality issues. There is also one reviewer zSUv has the methodology and novelty concern, where the reviewer point out pFedMoAP seems like a simple combination of PromptFL and MoE, lack of a strong well explained intuition. In the initial review, there are also criticism regarding the experimental gaps like varying the value of K, and concerns regarding privacy of directly downloading prompt from other clients and management challenges. While most of the concerning points have been addressed after the rebuttal, the issue of missing evaluations on large-scale benchmarks such as ImageNet, which have been used in other recent work, remains unresolved. The authors argue that this is due to a lack of time and resources. I still have some concerns about it. However, the most critical concerns regarding privacy and missing experiments have been addressed. After careful consideration of all the reviews, the authors’ response, and subsequent discussion, I am inclined to accept this paper.

**Additional Comments On Reviewer Discussion:**

The critical weaknesses in the initial review round were the lack of certain analysis experiments, concerns about privacy, and unclear presentation. While the authors have addressed most of these critical points by adding a differential privacy experiment, conducting additional experiments like on varying K,, and clarifying previously unclear points, the current version of the paper now presents a more convincing story. The remaining unresolved concern is the lack of experiments on large-scale datasets like ImageNet. Although I still have some reservations, considering that the sample size of ImageNet or UCF101 is not excessively large as well, I am ok to overlook this issue. Therefore, I tend to accept this paper.

---

### Decision · Program_Chairs · 2025-01-22

Accept (Poster)